# CONVERGENCE OF OPTIMIZERS IMPLIES EIGENVALUES FILTERING AT EQUILIBRIUM

## ABSTRACT

Ample empirical evidence in deep neural network training suggests that a variety of optimizers tend to find nearly global optima. In this article, we adopt the reversed perspective that convergence to an arbitrary point is assumed rather than proven, focusing on the consequences of this assumption. From this viewpoint, in line with recent advances on the edge-of-stability phenomenon, we argue that different optimizers effectively act as eigenvalue filters determined by their hyperparameters. Specifically, the standard gradient descent method inherently avoids the sharpest minima, whereas Sharpness-Aware Minimization (SAM) algorithms go even further by actively favoring wider basins. Inspired by these insights, we propose two novel algorithms that exhibit enhanced eigenvalue filtering, effectively promoting wider minima. Our theoretical analysis leverages a generalized Hadamard–Perron stable manifold theorem and applies to general semialgebraic $C^2$ functions, without requiring additional non-degeneracy conditions or global Lipschitz bound assumptions. We support our conclusions with numerical experiments on feed-forward neural networks.

## 1 INTRODUCTION

The stability of optimization algorithms has emerged as a key factor in understanding both the training dynamics and generalization of deep neural networks (Wu et al., 2018; Cohen et al., 2021). Stable training is correlated with desirable properties such as wide attraction basins and flat minima, which relate to generalization performances. This perspective underlies recent developments like sharpness-aware minimization (SAM) for finding flatter solutions (Foret et al., 2021), as well as empirical and practical analyses of the implicit bias of gradient methods toward low-curvature minima (Mulayoff et al., 2021). Moreover, the empirically observed "edge of stability" phenomenon for gradient descent and its variants (Cohen et al., 2021; 2022; Kaur et al., 2022; Andreyev & Ben-eventano, 2024) has highlighted that standard training often operates near the boundary of stability (e.g. learning rates are eventually close to the largest stable value). Theoretical results in dynamical systems and optimization further show that, under generic conditions, gradient-based algorithms avoid strict saddle points Lee et al. (2016); Panageas & Piliouras (2017); Ahn et al. (2022). Collectively, these observations indicate that stability plays a crucial role in where and how training converges (Ahn et al., 2022).

On the other hand, massive engineering efforts and improved heuristics over the past decade have made successful training almost the norm in deep learning practice, provided hyperparameters are well-tuned. This "systematic" convergence, and its tight link with the geometry of the loss landscape, invites a shift in perspective: rather than asking under what conditions an algorithm will converge, we ask *why a given successful training run did converge* and how the choice of hyperparameters made that possible.

In order to understand these phenomena in a unified way, we model optimizers by dynamics of the type

$$x_{k+1} = G_\alpha(x_k) = Dx_k - \alpha g(x_k) \qquad k = 0, 1, 2, \ldots, \tag{1}$$

where $D \in \mathbb{R}^{m \times m}$ is an invertible matrix, $g : \mathbb{R}^m \to \mathbb{R}^m$ is a $C^1$ continuously differentiable and semi-algebraic mapping (for example, $g$ could be the gradient of a loss function), and $\alpha > 0$ is a scalar step size. $D$ and $g$ may depend additionally on some fixed hyperparameters $p \in \mathbb{R}^\ell$; we single out the hyperparameter $\alpha$ to emphasize its role as the "learning rate", fundamental in practice. This formulation is quite general and covers many common optimization methods (gradient descent, heavy ball method, SAM) by an appropriate choice of $D$ and $g$.

As explained above we now focus on the *regime of successful runs* with a generalized form of the Hadamard–Perron stable manifold theorem:

**Theorem 1.1** (Successful runs imply nonexpansiveness at equilibrium). *Let $D \in \mathbb{R}^{m \times m}$ be an invertible matrix, $g : \mathbb{R}^m \to \mathbb{R}^m$ be a $C^1$ semi-algebraic mapping. For almost all $x_0 \in \mathbb{R}^m$ and $\alpha > 0$ the following assertion holds true: if the sequence $(x_k)_{k \in \mathbb{N}}$ converges to some point $\bar{x}$, then the spectral radius of the Jacobian of $D - \alpha g$ at $\bar{x}$ is at most $1$.*

With obvious notation, the conclusion reads $\rho\left(\operatorname{Jac} G_\alpha(\bar{x})\right) \leq 1$. This is a partial converse to the well-known local stability statement: when $\rho\left(\operatorname{Jac} G_\alpha(\bar{x})\right) < 1$, any sequence initialized sufficiently close to $\bar{x}$ converges. Now, if the dependence on hyperparameters is made explicit through $D = D(p)$, $g(x) = g(x, p)$, and if the dynamics is built upon the gradient of a loss $f$, then the inequality

$$\rho\left(\operatorname{Jac} G_\alpha(\bar{x}; p)\right) \;\leq\; 1$$

unfolds into "algebraic relations" tying the Hessian eigenvalues of $\nabla^2 f(\bar{x})$ to the hyperparameters $(\alpha, p)$. The fact that we are in a regime where convergence occurs shows the implicit effect that hyperparameters *filter limit points according to the eigenvalues of the Hessian*[1]. The edge-of-stability phenomenon is the empirical observation that the spectral radius bound tends to get saturated and in the long run hold rouhgly as an equality for deep neural network training (Cohen et al., 2021; 2022; Kaur et al., 2022; Andreyev & Beneventano, 2024).

As an intuition-building example, consider plain gradient descent (GD) on a $C^2$ loss. Classical local analysis around a nondegenerate minimum with Hessian eigenvalue $\lambda$ shows that convergence requires roughly $0 < \alpha < 2/\lambda$; otherwise, iterates oscillate or diverge. Conversely, if we place ourselves in a scenario where GD *does* converge, then necessarily $\lambda \leq 2/\alpha$ for all Hessian eigenvalues at the limit point. In other words, the reached points must satisfy the curvature bound $\lambda \leq 2/\alpha$: the method has *filtered out* points with higher curvatures. Whether convergence is guaranteed a priori, and to what extent this is realistic in full generality, remains open; nevertheless, deep learning offers a surprisingly rich empirical field that lends substantial support to the setting considered here (Cohen et al., 2021; 2022; Kaur et al., 2022; Andreyev & Beneventano, 2024). Note that Theorem 1.1 considerably extends the main result of Ahn et al. (2022) considered in this paragraph.

To illustrate our findings, we describe the relation between Hessian eigenvalues and hyperparameters for several optimization algorithms in terms of stable convergence: gradient descent, the heavy ball method, Nesterov's accelerated gradient method (with constant momentum). We push the investigation further in the context of sharpness aware minimization Foret et al. (2021), whose goal is to design recursive algorithms in the form of Equation (1) which tend to favor local minima with lower curvature. We focus on its un-normalized algorithmic variant, USAM Andriushchenko & Flammarion (2022); Dai et al. (2023), since the original SAM iteration are not smooth (not even continuous, due to normalization). Our analysis reveals that, the USAM algorithm induces more constraints on the limiting Hessian eigenvalues, which is consistent with the study of a simplified version of USAM in Zhou et al. (2025). To further illustrate this idea, we design two new SAM-based optimizer variants — *Two-step USAM* and *Hessian USAM* — which incorporate, respectively, an extra ascent step and second-order information into the SAM update. Our analysis predicts that these variants enforce *stricter* eigenvalue constraints under the convergence regime. We confirm these predictions empirically with numerical experiments on a multi-layer perceptron with MNIST and FASHION-MNIST datasets, as well as a wide ResNet architecture with the CIFAR10 dataset. These experiments qualitatively align with the prediction of the theory.

## 1.1 RELATED WORK

Anosov (1967) attributes the stable manifold theorem to Hadamard (1901) (see Hasselblatt & (Translator)) and Perron (1929) acknowledging earlier versions by Darboux, Poincaré and Lyapunov. Generic avoidance of strict saddle points by gradient flows dates back at least to Thom (1949). In an optimization context, similar ideas have been used for stochastic algorithms (Pemantle, 1990), inertial dynamics (Goudou & Munier, 2009), and more recently for the gradient algorithm in a machine learning context Lee et al. (2016); Panageas & Piliouras (2017); Ahn et al. (2022).

Implicit bias toward flat minima through stability is a common theme in the neural network literature (Wu et al., 2018; Ahn et al., 2022), with connection to generalization Mulayoff et al. (2021); Qiao et al. (2024); Wu et al. (2025); Kaur et al. (2022). This motivated the development of sharpness aware minimization algorithms (Foret et al., 2021; Andriushchenko & Flammarion, 2022; Dai et al., 2023) with several follow-up works on the connection between these approaches, flat minima, and prediction generalization (Andriushchenko et al., 2023; Marion & Chizat, 2024; Agarwala & Dauphin, 2023; Zhou et al., 2025; Tan et al., 2024)

The edge-of-stability phenomenon is the empirical observation that deep network training tends to saturate the "convergence stability constraints" on Hessian eigenvalues. This includes gradient descent (Cohen et al., 2021), adaptive methods (Cohen et al., 2022) and stochastic algorithms Andreyev & Beneventano (2024); Agarwala & Pennington (2024). This was studied theoretically for logistic regression (Wu et al., 2024; 2023) and in broader non convex optimization contexts (Damian et al., 2023; Arora et al., 2022; Ahn et al., 2022). The closest results to our main theorem are given in Lee et al. (2016); Panageas & Piliouras (2017); Ahn et al. (2022). We consider a much broader class of algorithms and under very mild hypotheses, effectively removing abstract non-degeneracy conditions or global Lipschitz bound assumptions.

---

[1]Note that we do not assume here any nondegeneracy of $\nabla f$ nor global Lipschitz properties at order 1.

## 2 LARGE STEP ANALYSIS OF ITERATIVE METHODS

Our main stability result is stated as follows.

**Theorem 2.1.** *Let $D \in \mathbb{R}^{m \times m}$ be an invertible matrix, $g \colon \mathbb{R}^m \to \mathbb{R}^m$ be a $C^1$ and semi-algebraic function and consider the recursion Equation (1). There exists $\Lambda \subset \mathbb{R}_+$, whose complement is finite, such that for any $\alpha \in \Lambda$, the set*

$$W_\alpha = \{x_0 \in \mathbb{R}^m \mid \exists \, \bar{x} \text{ s.t. } G_\alpha(\bar{x}) = \bar{x}, \, \rho(\mathrm{Jac}\, G_\alpha(\bar{x})) > 1, \, x_k \to \bar{x}, \, k \to \infty\}$$

*is contained in a countable union of $C^1$ submanifolds[2] of dimension at most $m - 1$.*

Since both finite sets and a countable union of $C^1$ lower-dimensional submanifolds have zero Lebesgue measure, one infers from Theorem 2.1 that for almost all $\alpha, x_0$, if the limit exists, then the corresponding spectral radius is at most 1. This form is Theorem 1.1 in the introduction, a result that we will use repeatedly. In Theorem 2.1, discarding a subset of step sizes and initializations is necessary as illustrated by the following example.

**Example 2.2.** Let $h \colon \mathbb{R} \to \mathbb{R}$ be $C^2$, such that $h(t) = (t^2 - 1)^2$ if $t \le 2$ and $h(t) = t^2$ if $t \ge 3$, and consider $f \colon \mathbb{R}^m \to \mathbb{R}$ such that $f(x) = h(\|x\|)$. The origin is a strict local maximizer such that $\nabla^2 f(0) = -4I$ ($I$ is the identity matrix with proper sizes). For any $\alpha > 0$, $\nabla f(0) = 0$ so that $0$ is fixed point of the gradient recursion with $\rho(\mathrm{Jac}\, G_\alpha(0)) = 1 + 4\alpha > 1$, hence $0 \in W_\alpha$. Furthermore for $\alpha = \frac{1}{2}$, $x - \alpha \nabla f(x) = 0$ for any $x$ such that $\|x\| \ge 3$. Hence $\{x \in \mathbb{R}^n \mid \|x\| \ge 3\} \subset W_\alpha$ and $W_\alpha$ is not as in Theorem 2.1.

Theorem 2.1 extends considerably (Ahn et al., 2022, Theorem 1), which was stated for the gradient with topological assumptions that we do not need. Moreover, our approach encompasses abstract dynamics and come with *easily verifiable assumptions*. Furthermore, the abstract form of Theorem 2.1 is more general. The proof of Theorem 2.1 leverages strong rigidity properties of semi-algebraic maps and a stable manifold theorem, as presented in the two following subsections. As stated in Remark 2.7 (b), we conjecture that a variant of Theorem 2.1 holds without the semi-algebraic assumption using Baire category arguments.

### 2.1 A STABLE MANIFOLD THEOREM BEYOND LOCAL DIFFEOMORPHISMS

Stability results similar to Theorem 2.1 are numerous (Pemantle, 1990; Goudou & Munier, 2009; Lee et al., 2016; Panageas & Piliouras, 2017; Ahn et al., 2022), they rely on variations of the Hadamard–Perron theorem. In dynamical systems theory, these are typically presented for local diffeomorphisms Hirsch et al. (1977); Shub et al. (1987). As presented in Example 2.2, being a local diffeomorphism may fail for general step size $\alpha$ as considered in Theorem 2.1.

It is actually known in dynamical systems literature that center stable manifold theorems hold beyond local diffeomorphisms, without requiring invertibility, as seen in the following result.

**Theorem 2.3** (Refined version of stable center manifold theorem). *Let $p$ be a fixed point for the $C^1$ function $F : U \to \mathbb{R}^n$ where $U \subseteq \mathbb{R}^n$ is an open neighborhood of $p$ in $\mathbb{R}^n$. Let $E_{sc} \oplus E_u$ be the invariant splitting of $\mathbb{R}^n$ into generalized eigenspaces of $\mathrm{Jac}\, F(p)$ corresponding to the eigenvalues of absolute value less or equal to 1, and strictly greater than 1 respectively. To the $\mathrm{Jac}\, F(p)$ invariant subspace $E_{sc}$, there is an associated local $F$ invariant $C^1$ submanifold $W_{loc}^{sc}$ of dimension $\dim(E_{sc})$, and a ball $B$ around $p$ such that:*

$$F(W_{loc}^{sc}) \cap B \subseteq W_{loc}^{sc},$$

*and if $F^k(x) \in B$ for all $k \ge 0$, then $x \in W_{loc}^{sc}$.*

Theorem 2.3 has already been presented or sketched in the literature and we provide a detailed and self-contained proof for completeness in Appendix B.

- In Shub et al. (1987) chapter 5, appendix III., there is a remark following the statement of Theorem III.2 to justify the existence of a center stable manifold when $F$ is a diffeomorphism. Then Exercise III.2 states that the invertibility of $F$ is actually not necessary.

- Similarly, in Hirsch et al. (1977), Theorem 5A.3 states a result about the existence of a center unstable manifold. A remark follows justifying the existence of a center stable manifold if $F$ is a diffeomorphism. The last paragraph of Section 5 from this book provides a quick justification of the fact that the invertibility of $F$ is not necessary.

### 2.2 SEMI-ALGEBRAICITY AND EXTENSION TO DEFINABLE OBJECTS

Theorem 1.1 is stated under semi-algebraic assumptions. The result probably holds beyond the semi-algebraic setting (see Remark 2.7 b), and use this as a versatile sufficient condition. We refer to Coste (2000a;b) for an introductory exposition of semi-algebraic and definable geometry as well as Attouch et al. (2010; 2013) for numerous examples in optimization.

---

[2]Without further precisions, in the main text, all submanifolds are supposed to be embedded.

**Definition 2.4** (Semi-algebraic sets and functions). A basic semi-algebraic subset of $\mathbb{R}^m$ is the solution set of a system of finitely many polynomial inequalities and a semi-algebraic subset is a finite union of basic semi-algebraic subsets. A semi-algebraic function is a function whose graph is semi-algebraic.

**Example 2.5** (Semi-algebraic functions). Affine, polynomial, rational, square root, relu, matrix rank, $\ell_p$ norms, maximum coordinate, argmax coordinate, sort operation ...

The composition of two semi-algebraic functions is semi-algebraic. For this reason the class of semi-algebraic functions is very well adapted to study deep networks, which are parameterized compositions. The training loss of a deep network built with semi-algebraic operations is semi-algebraic, *e.g.* a relu multilayer perceptron with the squared loss.

From a geometric point of view, semi-algebraicity ensures a form of rigidity. The following describes the main feature of semi-algebraic functions used in Theorem 2.1. It states that, apart from a finite number of step sizes, being a smooth manifold, a "small" subset, is preserved by the inverse of the algorithmic recursion Equation (1), up to countable unions. The proof is postponed to Appendix A. The proof of Theorem 2.1 globalizes the local stability result in Theorem 2.3 using Lemma 2.6 and standard arguments.

**Lemma 2.6.** *Let $D \in \mathbb{R}^{m \times m}$ be an invertible matrix, $g \colon \mathbb{R}^m \to \mathbb{R}^m$ be a $C^1$ and semi-algebraic function. Consider the function $G_\alpha$ defined as in Equation (1). There exists a subset $\Lambda \subseteq \mathbb{R}_{>0}$, whose complement is finite, such that for any $\alpha \in \Lambda$: if $S \subset \mathbb{R}^m$ is a $C^1$ submanifold of dimension at most $m-1$, the pre-image $G_\alpha^{-1}(S)$ is contained in a countable union of $C^1$ manifolds of dimension at most $m-1$.*

**Remark 2.7** (Beyond semi-algebraicity). (a) Definable case. Many deep learning losses involve the logarithm or exponential functions which are not semi-algebraic. There is a larger function class, i.e., functions definable in a certain o-minimal structure van den Dries & Miller (1996), which contains the logarithm, the exponential as well as all restrictions of analytic functions to compact balls in their domain. This class retains all the features we use to prove lemma 2.6. We state Lemma 2.6 and Theorem 2.1 for semi-algebraic functions for simplicity, but they actually holds for a much broader class and are applicable to virtually all smooth losses arising in deep learning. (b) Beyond definability. Our globalization strategy relies on the semi-algebraic (or definable) assumption, and we conjecture that a more general variation could be considered without the semi-algebraic assumption. In particular the subset $\Lambda$ in Lemma 2.6 is constructed by applying the Morse-Sard Theorem to the eigenvalues of $\text{Jac } g$. Since eigenvalues are Lipschitz, one could consider Lipschitz versions of the Morse-Sard Theorem, see *e.g.* the remark following (Evans & Gariepy, 2015, Theorem 3.1).

## 3 APPLICATIONS TO OPTIMIZATION ALGORITHMS

In the following, we apply Theorem 2.1 to multiple optimization algorithms; we use the "almost every" formulation of the introduction for simplicity. Technical details are postponed to Appendix C

### 3.1 CAUCHY'S GRADIENT DESCENT

The update rule of gradient descent (GD) is given by:

$$x_{k+1} = x_k - \alpha \nabla f(x_k). \tag{2}$$

**Proposition 3.1** (Gradient descent eigenvalues filtering). *Assume that $f$ is $C^2$ and semi-algebraic. For almost every $\alpha > 0$ and $x_0 \in \mathbb{R}^n$, we have: if $\{x_k\}_{k \in \mathbb{N}}$ given by Equation (2) converges to $\bar{x}$, then all eigenvalues $\lambda$ of $\nabla^2 f(\bar{x})$ satisfy: $0 \le \lambda \le \frac{2}{\alpha}$.*

The stability condition for this algorithm is very well known, let us compare Proposition 3.1 with existing results of the same kind.

*Avoidance of strict saddle points:* The conclusion $\lambda \ge 0$ illustrates the well known fact that gradient descent escapes strict saddle points. This result has already been stated multiple times in many different forms (e.g., Thom (1949); Goudou & Munier (2009); Lee et al. (2016); Panageas & Piliouras (2017)). In particular, (Pemantle, 1990, Theorem 1) applies to stochastic gradient descent with vanishing step-sizes, (Lee et al., 2016, Theorem 4.1) and (Panageas & Piliouras, 2017, Theorems 2,3) applies to Lipschitz gradients in the stable, small step, regime. In comparison Proposition 3.1 requires minimal qualitative assumptions on $f$ and is valid for a much broader range of step sizes.

*Large step sizes and small curvature:* the upper-bound $\lambda \le 2/\alpha$ is a core element of the Edge Of Stability (EOS) phenomenon Cohen et al. (2021), crucial for understanding training dynamics. Most often in the EOS literature, stability mechanisms are justified on quadratic objectives, for which computation is very simple. Our result shows that these conclusions extend to a generic deep learning setting. Proposition 3.1 is similar to (Ahn et al., 2022, Theorem 1) without the need for the abstract (Ahn et al., 2022, Assumption 1), for a generic step-size.

### 3.2 POLYAK'S HEAVY BALL METHOD

The method's iterations update is given by:

$$\begin{pmatrix} x_{k+1} \\ y_{k+1} \end{pmatrix} = \underbrace{\begin{pmatrix} (1+\beta)I & -\beta I \\ I & 0_{n\times n} \end{pmatrix}}_{D} \begin{pmatrix} x_k \\ y_k \end{pmatrix} - \alpha \underbrace{\begin{pmatrix} \nabla f(x_k) \\ 0 \end{pmatrix}}_{g(\cdot)}, \tag{3}$$

where $I$ is the identity matrix and $0_{n\times n}$ is an all-zero one. Considering Theorem 2.1, the eigenvalue computation is also known and was carried out for example in Polyak (1987).

**Proposition 3.2** (Heavy Ball eigenvalues filtering). *Assume that $f$ is $C^2$, semi-algebraic and $0 < \beta < 1$ in Equation* (3). *For almost every $\alpha > 0$ and $(x_0, y_0) \in \mathbb{R}^n \times \mathbb{R}^n$, we have: if $\{(x_k, y_k)\}_{k\in\mathbb{N}}$ converges to some $(\bar{x}, \bar{y})$, then all the eigenvalues $\lambda$ of $\nabla^2 f(\bar{x})$ satisfy:*

$$0 \leq \lambda \leq \frac{2(1+\beta)}{\alpha}.$$

We review existing results related to Proposition 3.1:

*Avoidance of strict saddle points and generic convergence to minimizers:* for the heavy ball method, this was documented for the continuous time ODE limit of the method Goudou & Munier (2009), and in discrete time for Lipschitz gradients under small step size conditions Sun et al. (2019); Castera (2021). Our result holds for generic step-sizes.

*Large step sizes and small curvature:* if the iterates of Equation (3) converges, the limiting curvature is upper bounded by $2(1 + \beta)/\alpha$. This upper bound appeared in (Cohen et al., 2021, Equation 1, Theorem 2) for the quadratic case and we extend it to a general setting.

### 3.3 NESTEROV ACCELERATED GRADIENT METHOD

We further illustrate our result with Nesterov's accelerated gradient method (NAG) with fixed momentum constant $\beta$, the original version being described with decaying $\beta$ Nesterov (1983). Fixed momentum, however, is used frequently in the deep learning context, for example this corresponds to the Pytorch implementation. Its iteration update is given by:

$$\begin{pmatrix} x_{k+1} \\ y_{k+1} \end{pmatrix} = \underbrace{\begin{pmatrix} (1+\beta)I & -\beta I \\ I & 0_{n\times n} \end{pmatrix}}_{D} \begin{pmatrix} x_k \\ y_k \end{pmatrix} - \alpha \underbrace{\begin{pmatrix} \nabla f(x_k + \beta(x_k - y_k)) \\ 0 \end{pmatrix}}_{g(\cdot)} \tag{4}$$

**Proposition 3.3** ("Nesterov's accelerated gradient"[3]). *Assume that $f$ is $C^2$, semi-algebraic and $0 < \beta < 1$ in Equation* (3). *For almost every $\alpha > 0$ and $(x_0, y_0) \in \mathbb{R}^n \times \mathbb{R}^n$, we have: if $\{(x_k, y_k)\}_{k\in\mathbb{N}}$ converges to some $(\bar{x}, \bar{y})$ , then all the eigenvalues $\lambda$ of $\nabla^2 f(\bar{x})$ satisfy:*

$$0 \leq \lambda \leq \frac{1}{\alpha}\left(\frac{2+2\beta}{1+2\beta}\right).$$

An existing line of works Jin et al. (2018); Agarwal et al. (2017); Carmon et al. (2018) exploits the ideas of Nesterov's method to investigate the complexity of finding approximate second order critical points. The nature of our results is different: the method generically avoids strict saddle points and filters eigenvalues more strongly as $\beta$ grows. In addition, the upper bound on eigenvalues was described for the quadratic case (Cohen et al., 2021, Appendix B).

*Generic convergence and eigenvalues filtering*: This algorithm shares the property of generic convergence to minimizers with the gradient descent (2), and the Heavy Ball method (3). Observe that its filtering abilities are slightly improved over the gradient descent ($2/\alpha$) and the Heavy Ball method ($4/\alpha$), as the eigenvalue upper bound tends to $4/(3\alpha)$ when $\beta$ approaches 1.

### 3.4 UNNORMALIZED SHARPNESS AWARE MINIMIZATION (USAM)

USAM was introduced and studied in Andriushchenko & Flammarion (2022). Its iteration update is given by:

$$x_{k+1} = x_k - \alpha \nabla f(x_k + \rho \nabla f(x_k)) \tag{5}$$

for some constant $\rho > 0$. This is a modified version of the original SAM Foret et al. (2021), given by:

$$x_{k+1} = x_k - \alpha \nabla f\left(x_k + \rho\frac{\nabla f(x_k)}{\|\nabla f(x_k)\|}\right) \tag{6}$$

---

[3]Here we adopt the ML community's terminology: 'accelerated' refers to the ideal case when the loss is strongly convex.

We focus on Equation (5) because the update rule of Equation (6) is not $C^1$, it is actually discontinuous around critical points. USAM is a modified version of gradient descent, $\nabla f(x_k)$ being simply replaced by $\nabla f(x_k + \rho \nabla f(x_k))$. In practice, this is combined with other techniques such as heavy ball momentum as follows:

$$x_{k+1} = x_k + \beta(x_k - x_{k-1}) - \alpha \nabla f(x_k + \rho \nabla f(x_k)) \tag{7}$$

or equivalently,

$$\begin{pmatrix} x_{k+1} \\ y_{k+1} \end{pmatrix} = \underbrace{\begin{pmatrix} (1+\beta)I & -\beta I \\ I & 0_{n \times n} \end{pmatrix}}_{D} \begin{pmatrix} x_k \\ y_k \end{pmatrix} - \alpha \underbrace{\begin{pmatrix} \nabla f(x_k + \rho \nabla f(x_k)) \\ \mathbf{0} \end{pmatrix}}_{g(\cdot)} \tag{8}$$

to conform with the standard form of Equation (1).

**Proposition 3.4** (USAM + Heavy Ball momentum). *Assume that $f$ is $C^2$, semi-algebraic and $0 \leq \beta < 1$ in Equation (7). For almost every $\alpha > 0$ and $(x_0, y_0) \in \mathbb{R}^n \times \mathbb{R}^n$, we have: if $\{(x_k, y_k)\}_{k \in \mathbb{N}}$ converges to some $(\bar{x}, \bar{y})$ where $\nabla f(\bar{x}) = 0$, then all the eigenvalues $\lambda$ of $\nabla^2 f(\bar{x})$ satisfy:*

$$0 \leq \lambda(1 + \rho\lambda) \leq \frac{2(1+\beta)}{\alpha}.$$

*or equivalently,*

$$\frac{-1 - \sqrt{1 + 8(1+\beta)\rho/\alpha}}{2\rho} \leq \lambda \leq -\frac{1}{\rho} \quad or \quad 0 \leq \lambda \leq \frac{\sqrt{1 + 8(1+\beta)\rho/\alpha} - 1}{2\rho}.$$

Let us discuss the implications Proposition 3.4 in light of existing literature:

*Strict saddle points may be attractive*: USAM with or without momentum does not avoid strict saddle points generically. For example, if $f(x) = -\frac{1}{\rho}x^2$ and $0 < \alpha < \rho$, then $(0,0)$ is a stable fixed point since one can prove that the update in Equation (8) is locally a contraction at $(0,0)$. This was remarked in (Kim et al., 2023, Theorem 1) for the ODE version of Equation (5) and is seen in Proposition 3.4 with the negative interval.

*Apparition of new fixed points*: The set of fixed points of the USAM dynamics in Equation (7) is given by

$$\{x : \ x + \rho \nabla f(x) \in \mathrm{crit} f\} \times \{0\} \supset \mathrm{crit} f \times \{0\}.$$

Thus, the set of fixed points of the USAM algorithm is possibly strictly larger than the set of critical points of the underlying loss function. As shown in Appendix D, the set of fixed points of the USAM algorithm not belonging to $\mathrm{crit} f$ may even have a nonempty interior. This remark combined with the previous one on strict saddle points illustrate that, in full generality, the USAM algorithm does not enjoy the property of generic convergence to local minizers of the objective $f$, contrary to the gradient or heavy ball algorithms. In the context of deep learning, we observe however that the USAM algorithm has a minimizing behavior, suggesting that the spurious fixed points have a limited impact.

*Eigenvalues filtering*: The upper bound in Proposition 3.4 is smaller than that of Proposition 3.1 for any $\rho > 0$. Both the upper and lower bounds are of order $1/\sqrt{\alpha\rho}$ as $\rho \to \infty$. The upper bound in Proposition 3.4 was described in Zhou et al. (2025) for a simplified version of USAM. Moreover, for a fixed $\rho > 0$, as $\alpha \to 0$, these bounds scale like $O(1/\sqrt{\alpha})$, while for previous methods the upper bound scales like $O(1/\alpha)$. These observations suggest that USAM may converge to flatter critical points of the objective.

### 3.5 Two variants of USAM finding flat minimizers

We investigate two variations on USAM which result in finer constraints on asymptotic curvature. For both, for a fixed SAM parameter $\rho$, the upper bound scales like $\alpha^{-1/3}$ as $\alpha \to 0$, which is smaller the one found for USAM.

#### 3.5.1 Two-step USAM

The following update performs two gradient ascent steps:

$$x_{k+1} = x_k - \alpha \nabla f(\underbrace{x_k + \rho \nabla f(x_k) + \rho \nabla f(x_k + \rho \nabla f(x_k))}_{\text{two gradient ascent steps}}) \tag{9}$$

for some constant $\rho > 0$. Combining with heavy ball momentum gives:

$$\begin{pmatrix} x_{k+1} \\ y_{k+1} \end{pmatrix} = \begin{pmatrix} (1+\beta)I & -\beta I \\ I & 0_{n \times n} \end{pmatrix} \begin{pmatrix} x_k \\ y_k \end{pmatrix} - \alpha \begin{pmatrix} \nabla f(x_k + \rho \nabla f(x_k) + \rho \nabla f(x_k + \rho \nabla f(x_k))) \\ \mathbf{0} \end{pmatrix} \tag{10}$$

**Proposition 3.5** (Two-step USAM gradient). *Assume that $f$ is $C^2$, semi-algebraic, $\rho > 0$ and $0 \leq \beta < 1$ in Equation (10). For almost every $\alpha > 0$ and $(x_0, y_0) \in \mathbb{R}^n \times \mathbb{R}^n$, we have: if $\{(x_k, y_k)\}_{k \in \mathbb{N}}$ converges to $(\bar{x}, \bar{y})$ where $\bar{x}$ is a critical point of $f$, i.e., $\nabla f(\bar{x}) = 0$, then all the eigenvalues of $\lambda$ of $\nabla^2 f(\bar{x})$ satisfy:*

$$0 \leq \lambda(1 + \rho\lambda)^2 \leq \frac{2(1+\beta)}{\alpha}.$$

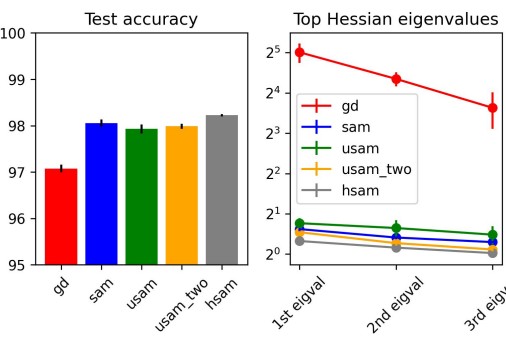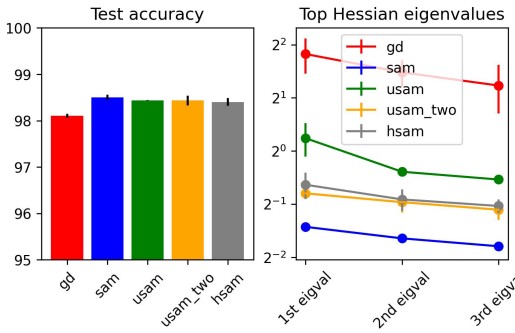

**Figure 1:** (**Experiment 1**) - MLP trained on MNIST with *stochastic gradient descent* and its corresponding to SAM, USAM, USAM2 and Hessian USAM versions. Left without momentum, right with $\beta = 0.9$. SAM, USAM and USAM2 are trained with $\rho \in \{0.05, 0.1, 0.2\}$ while Hessian USAM is trained with $\rho \in \{0.01, 0.02, 0.05, 0.1, 0.2\}$. Among these $\rho$, we choose those yielding the best models (in terms of test accuracy) and report their accuracy and hessian spectra.

**Remark 3.6** (Convergence & improved eigenvalues filtering). Contrary to USAM, Two-step USAM avoids strict saddle points generically as the interval given in Proposition 3.5 does not allow for negative $\lambda$. Yet the nonempty interior argument of Appendix D maybe adapted and, for simple costs, the algorithm may have many spurious fixed points, which do not correspond to critical points of the objective. As for USAM, empirical results suggest that they have a limited effect on the minimizing behavior of the algorithm in deep learning.

As for eigenvalue filtering, the result is rather positive; if USAM and two-step USAM have the same common hyperparameters, we infer that, within $\text{crit} f$, stable fixed points for Two-step USAM are also stable for USAM (the converse being not necessarily true). This suggests that Two-step USAM can find flatter local minima.

### 3.5.2 HESSIAN USAM

We consider the following iteration update:

$$x_{k+1} = x_k - \alpha \nabla f(x_k + \rho \nabla^2 f(x_k) \nabla f(x_k)), \tag{11}$$

replacing $\nabla f(x_k)$ as in Equation (2) by $\nabla f(x_k + \rho \nabla^2 f(x_k) \nabla f(x_k))$. Combining with heavy ball momentum gives:

$$\begin{pmatrix} x_{k+1} \\ y_{k+1} \end{pmatrix} = \begin{pmatrix} (1+\beta)I & -\beta I \\ I & 0_{n \times n} \end{pmatrix} \begin{pmatrix} x_k \\ y_k \end{pmatrix} - \alpha \begin{pmatrix} \nabla f(x_k + \rho \nabla^2 f(x_k) \nabla f(x_k)) \\ \mathbf{0} \end{pmatrix} \tag{12}$$

**Proposition 3.7** (Hessian USAM gradient). *Assume that $f$ is $C^2$, semi-algebraic, $\rho > 0$ and $0 \leq \beta < 1$ in Equation (10). For almost every $\alpha > 0$ and $(x_0, y_0) \in \mathbb{R}^n \times \mathbb{R}^n$, we have: if $\{(x_k, y_k)\}_{k \in \mathbb{N}}$ converges to $(\bar{x}, \bar{y})$ where $\bar{x}$ is a critical point of $f$, i.e., $\nabla f(\bar{x}) = 0$, then all the eigenvalues of $\lambda$ of $\nabla^2 f(\bar{x})$ satisfy:*

$$0 \leq \lambda(1 + \rho\lambda^2) \leq \frac{2(1+\beta)}{\alpha}.$$

**Remark 3.8** (Convergence & improved eigenvalues filtering). Like Two-step USAM (Remark 3.6), Hessian USAM avoids strict saddle points. However, it may also fail to achieve generic convergence to local minimizers because spurious fixed points may generate stable points out of $\text{crit} f$, recall Appendix D. Once again, its eigenvalue-filtering properties are generally improved over USAM.

## 4 EXPERIMENTS

In this section, we evaluate numerically limiting curvature at equilibrium for the considered algorithms, in the context of neural networks training. In our experiments, we compare in particular the popular (stochastic) gradient descent and heavy-ball method with their corresponding USAM, Two-step USAM, and HSAM versions to observe the effect of eigenvalues filtering. We do not implement the Nesterov algorithm and its SAM variants since they are less commonly used in the neural networks training context.

We conducted three neural network training experiments described below; our Python implementation is available at `a_public_Github_repo_after_anonymous_review` for reproduction purposes. The datasets, architectures and protocols are as follows:

1. **MNIST dataset and MultiLayer Perceptron (MLP)**: The dimensions of hidden layers are $\{128, 64, 10, 10\}$, with ReLU activation function. We use the standard cross-entropy loss for classification.

   We consider GD, SAM, USAM, USAM2 and Hessian SAM with ($\beta = 0.9$) and without ($\beta = 0$) momentum. We fix a 128 minibatch size, an $\alpha = 0.01$ learning rate a weight decay of $5e - 4$. The parameter $\rho$ is tuned from

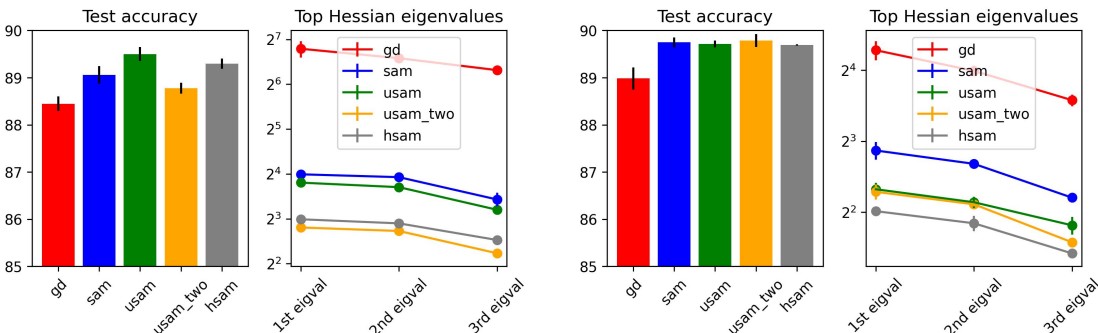

Figure 2: (**Experiment 2**) - Same as Figure 1 with the MNIST-FASHION dataset.

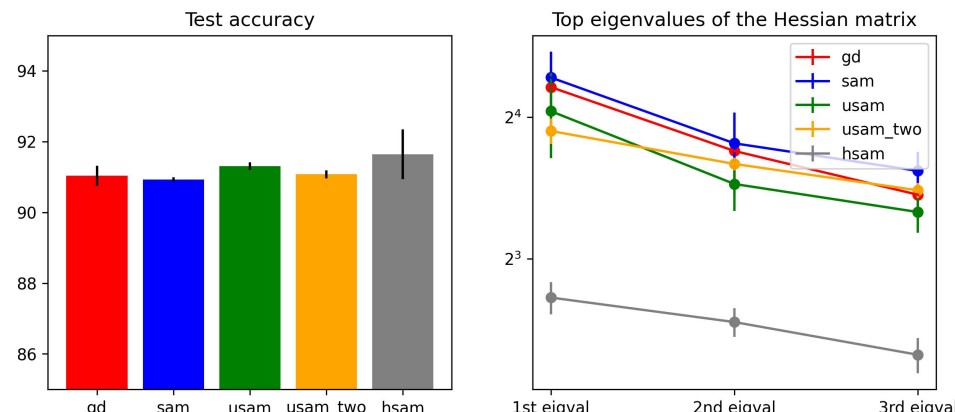

Figure 3: (**Experiment 3**) - Models are trained with *stochastic gradient descent* and its corresponding to SAM, USAM, USAM2 and Hessian USAM versions. The values of $\rho$ of all SAM-like algorithms are set at $\rho = 0.001$.

grid search: for SAM variants, we tune the hyperparameter $\rho \in \{10^{-i}, 2 \times 10^{-i}, 5 \times 10^{-i} \mid i \geq 1, i \in \mathbb{N}\}$. We consider the best run in terms of accuracy among the three largest values of $\rho$ such that the training does not fail. We report the corresponding three largest eigenvalues of the Hessian matrix of the training loss after training. The training is repeated three times for each set of hyperparameters. The results are illustrated in Figure 1.

2. **MNIST-fashion dataset and MultiLayer Perceptron (MLP)**: We use the same setting as in the first experiment, except replacing the MNIST dataset with the MNIST-fashion dataset Xiao et al. (2017). The results are illustrated in Figure 2.

3. **CIFAR10 dataset and WideResNet-16-8**: The third experiment consists of training a WideResNet-16-8 (Zagoruyko & Komodakis (2016)) without batch normalization layers with CIFAR10 dataset. This specification echoes the remark from (Foret et al., 2021, Section 4.2) whose authors suggest that batch normalization layers tend to "obscure interpretation of the Hessian". Our choice of WideResNet architecture is motivated by previous experiments reporting succesful training without batch normalization. We consider the momentum version of GD, SAM, USAM, USAM2 and Hessian SAM with the same value of $\rho = 0.001$. The results are shown in Figure 3.

The MLP experiments illustrate the fact that our theoretical findings transfer well to the experimental setting: in general, methods such as USAM, USAM2 and Hessian USAM consistently find flatter (or low-curvatured) minimizers in comparison to their vanilla versions. The experiment is far from the idealized assumptions in Theorem 2.1, with non-smoothness (since we use ReLU neural networks) and stochasticity (in the optimization algorithms), but the theory definitely aligns with empirical results: USAM2 and Hessian USAM filter Hessian eigenvalues more efficiently, and find therefore solutions with wider basins.

As for the WideResNet the situation is not as clear in Figure 3, the differences are less pronounced, notably between USAM and USAM2. This architecture is much more difficult to train than the MLP architecture considered above. Another outcome of the experiment, which aligns well with our stability analysis is as follows. For a fixed $\alpha$, USAM2 and Hessian USAM require much smaller values of $\rho$ than USAM to avoid training failure. This is consistent with the fact that both USAM2 and Hessian USAM enforce more restrictions on the limiting curvature

in comparison to USAM. For this reason we needed to significantly decrease the value to $\rho = 0.001$ in order to obtain successful training. In this regime, the reduction of asymptotic curvature is limited.

Note that SAM also consistently finds flat minimizers, this fact is experimentally confirmed by previous works Foret et al. (2021); Tan et al. (2024) in several deep learning settings. Our empirical result resonates with these observations. Nevertheless, our theoretical results do not provide any explanation for this behavior and we leave this extension to future work.

Our last remark is that with architectures using batch normalization, SAM (and also USAM, USAM2 and Hessian SAM) does not empirically converge to flatter minimizers (see (Foret et al., 2021, Section 4.2)). This does not contradict our theoretical result because the presence of batch normalization changes the dynamics of the algorithm. Another future direction is to extend Theorem 2.1 to also cover batch normalization operations.

## 5 CONCLUSION

We provide a simple, general, and versatile theoretical result on eigenvalues filtering (cf. Theorem 2.1) in the context of convergence of optimization algorithms. This takes the form of a variation on the Hadamard–Perron stable manifold theorem, which simplifies and generalizes existing results of this type in the machine learning literature. The proposed result aligns with recent empirical and theoretical advances in sharpness-aware minimization, large step size, generalization, and edge-of-stability phenomena.

We introduced two new algorithms, Two-step USAM and Hessian SAM. These algorithms are given to illustrate our theoretical findings on algorithmic stability (Theorem 2.1) in a deep network training scenario. We emphasize that the computational cost of a single iteration for each algorithm is higher than that of USAM. For this reason, we do not have empirical evidence that Two-step USAM or Hessian SAM provides a substantially practical advantage compared to SAM or USAM. They nonetheless illustrate the generality of the proposed theoretical analysis, and we leave extensive benchmarking of their empirical performance to future work.

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

## A    PROOF OF THEOREM 2.1

We first provide preliminary lemmas and then proceed to the proof of Lemma 2.6 and Theorem 2.1.

**Lemma A.1.** *Let $F : \mathbb{R}^m \to \mathbb{R}^m$ be a semi-algebraic function and $S \subset \mathbb{R}^m$ be a $C^1$ embedded submanifold of dimension at most $m - 1$. Then, $F(S)$ is contained in a countable union of $C^1$ embedded submanifolds of dimension at most $m - 1$.*

*Proof.* Invoking (van den Dries & Miller, 1996, Lemma C.2) there exists $P_1, \ldots, P_N$, disjoint, semi-algebraic and open, whose union is dense in $\mathbb{R}^m$, such that for each $k = 1, \ldots, N$, the restriction $F_k = F \mid_{P_k} : P_k \to \mathbb{R}^m$ is $C^1$ and $J_k := \text{Jac } F_k$ has constant rank. Set $P_0 = \cap_{k=1}^N P_k^c$, $P_0$ is semi-algebraic of dimension at most $m - 1$. Partition the set $S$ into:
$$S_k = P_k \cap S, \qquad \forall k = 0, 1, \ldots, N.$$
We have that $\cup_{k=0}^N P_k = \mathbb{R}^m$ so that $F(S) = \cup_{k=0}^m F_k(S_k)$. Consider three cases:

1. $k = 0$: $F(S_0) \subset F(P_0)$ which is of dimension at most $m - 1$ by (van den Dries & Miller, 1996, 4.7) so that $F(P_0)$ is contained in a finite union of $C^1$ embedded submanifolds of dimension at most $m - 1$ (van den Dries & Miller, 1996, 4.8).

2. $k > 0$ and $\text{rank}(J_k) < m$: Then $S_k \subseteq P_k$ is a subset of the critical points of the mapping $F$. Since $F_k : P_k \to \mathbb{R}^m$ is $C^1$, we can apply the Sard theorem to conclude that $\mu(F(P_k)) = 0$ ($\mu(\cdot)$ is the Lebesgue measure). Since $F(P_k)$ is also semi-algebraic (because $F$ and $P_k$ are semi-algebraic), its zero Lebesgue measure implies that $F(S_k) \subseteq F(P_k)$ is contained in a finite union of $C^1$ embedded submanifolds.

3. If $k > 0$ and $\text{rank}(J_k) = m$: then $F_k$ is a local diffeomorphism. It implies that for any $x \in S$, there exists an open neighborhood $V_x \subseteq P_k$ such that $F(S \cap V_x)$ is an embedded submanifold of dimension at most $m - 1$. By taking a countable open covering $\{V_i, i \in \mathbb{N}\}$ of $S_k$, we have:
$$F_k(S_k) \subseteq \bigcup_{i \in \mathbb{N}} F(S \cap V_i),$$

and hence, the result. $\qquad\square$

**Lemma A.2.** *Consider a semi-algebraic set $S \subset \mathbb{R}^n \times \mathbb{R}^m$. If for all $x \in \mathbb{R}^n$, the fiber $S_x = S \cap \{x\} \times \mathbb{R}^m$ only contains isolated points. Then there exists an integer $N$ and $N$ semi-algebraic functions $F_1, \ldots, F_N : \text{proj}_{\mathbb{R}^n} S \to \mathbb{R}^m$, $k = 1, \ldots, N$ such that:*
$$S = \bigcup_{k=1}^N \text{graph } F_k. \tag{13}$$

*Proof.* By (van den Dries & Miller, 1996, Properties 4.4), the number of connected components of $S_x$ is uniformly bounded by a number $N \in \mathbb{N}$. Moreover, the connected components are singletons because they only consist of isolated points. Therefore, there exists a positive integer $N$ such that $|S_x| < N$, for all $x \in \mathbb{R}^n$.

We construct the semi-algebraic functions $F_1, \ldots, F_N$ recursively as follows. Set $S_1 = S$ by (van den Dries & Miller, 1996, Property 4.5), there is a semi-algebraic function $F_1 : \text{proj}_{\mathbb{R}^n} S_1 \to \mathbb{R}^m$ such that $\text{graph } F_1 \subset S_1 = S$. We set $\mathcal{D} = \text{proj}_{\mathbb{R}^n} S$, the domain of $F_1$. Recursively, we set for $k \geq 2$, $S_k = S_{k-1} \setminus \text{graph } F_{k-1}$ and define similarly $F_k : \text{proj}_{\mathbb{R}^n} S_k \to \mathbb{R}^m$ such that $\text{graph } F_k \subset S_k \subset S$. At each iterations the cardinality of the fibers of $S_k$ is reduced by at least 1 compared to those of $S_{k-1}$. After $N$ iterations, we have $S_N \setminus \text{graph } F_N = \emptyset$ and $S = \bigcup_{k=1}^N \text{graph } F_k$. Each function can be extended to the whole set $\mathcal{D}$ by choosing the value $F_k(x) = F_1(x)$ outside of the domain of definition of $F_k$. This preserves semi-algebraicity as well as the equality; this concludes the proof. $\qquad\square$

*Proof of Lemma 2.6.* It is sufficient to prove the result for $D = I$. Indeed, we can rewrite $G_\alpha = D(x - \alpha D^{-1} g(x))$. If we find $\Lambda$ satisfy Lemma 2.6 for $\bar{G}_\alpha := x - \alpha D^{-1} g(x)$, then the same $\Lambda$ also works for $G_\alpha$ since $G_\alpha$ is a composition of a global diffeomorphism $x \mapsto Dx$ and $\bar{G}_\alpha$. Therefore, in the following, we can assume that $D = I$, i.e. $G_\alpha(x) = x - \alpha g(x)$.

Consider $\lambda^{\mathcal{R}}(x) : \mathbb{R}^m \to \mathbb{R}^m : (\lambda_i^{\mathcal{R}})_{i=1}^m$, the real parts of eigenvalues of the Jacobian matrix $\text{Jac } g(x)$, counted with their multiplicity. Since $\lambda_i^{\mathcal{R}}, i = 1, \ldots, m$ are semi-algebraic, we choose a semi-algebraic, open, and dense subset $I \subset \mathbb{R}^m$ so that $\lambda_i^{\mathcal{R}}, i = 1, \ldots, M$ are all differentiable on $I$.

We define $\Lambda := \{\alpha > 0 \mid \alpha^{-1} \notin \cup_{i=1}^m \lambda_i^{\mathcal{R}}(\text{crit}\lambda_i^{\mathcal{R}})\}$. We prove that $\Lambda$ satisfies the conditions of Lemma 2.6. Due to the semi-algebraic Sard's theorem Kurdyka et al. (2000), the set of critical values of $\lambda_i^{\mathcal{R}}$ is of zero Lebesgue

measure. Moreover, it is also semi-algebraic. Hence, $\cup_{i=1}^m \lambda_i^{\mathcal{R}}(\text{crit}\lambda_i^{\mathcal{R}})$ is finite. Thus, the complement $\mathbb{R}_{>0} \setminus \Lambda$ is finite. Fix $\alpha \notin \Lambda$, we are going to verify the required condition.

Set $K_\alpha = \{x \in \mathbb{R}^m \mid \det(I - \alpha\text{Jac } g(x)) = 0\}$, which is semi-algebraic. We are going to show that $\dim(K_\alpha) < m$. Partition $K_\alpha$ into two sets:

$$K_1 = K_\alpha \cap I \qquad \text{and} \qquad K_2 = K_\alpha \cap I^c,$$

where $I$ is a semi-algebraic dense open set in which all functions $\lambda_i^{\mathcal{R}}, i = 1, \ldots, m$ are differentiable. Since $I$ is open, dense and semi-algebraic, $\dim(I^c) < m$ and thus, $\dim(K_2) < m$. To prove that $\dim(K_1) < m$, we notice that:

$$K_\alpha \subseteq \cup_{i=1}^m K_{\alpha,i} \quad \text{where} \quad K_{\alpha,i} := \{x \in I \mid \alpha\lambda_i^{\mathcal{R}}(x) = 1\}.$$

The sets $K_{\alpha,i}, i = 1, \ldots, m$ are semi-algebraic themselves. Their dimension has to be strictly smaller than $m$. Indeed, by contradiction, if there exists $i$ such that $K_{\alpha,i}$ has dimension $m$, it must contain an open set $O$. For all $x \in O$, $\lambda_i^{\mathcal{R}}(x) = 1/\alpha$ is a constant. Therefore, $1/\alpha$ is a critical value of $\lambda_i^{\mathcal{R}}$, which contradicts our choice of $\alpha$. Thus, $K_{\alpha,i}$ and $K_\alpha$ have dimension strictly smaller $m$.

Given an embedded $C^1$ submanifold $S$ of dimension $p < m$, we partition its pre-image by $G_\alpha, \alpha \in \Lambda$ as:

$$S_1 = G_\alpha^{-1}(S) \cap K_\alpha \qquad \text{and} \qquad S_2 = G_\alpha^{-1}(S) \cap K_\alpha^c.$$

Since $K_\alpha$ is semi-algebraic of dimension at most $m - 1$, it is contained in a finite union of submanifolds of dimension at most $m - 1$. It is sufficient to show that the same property holds for $S_2$ as well. Consider the following set:

$$T = \{(y, x) \mid y = x - \alpha g(x) \text{ and } x \in K_\alpha^c\} \subseteq \mathbb{R}^{2m}.$$

$S_2$ is the projection of $T \cap S \times \mathbb{R}^m$ onto the last $m$ coordinates. The set $T$ is semi-algebraic (although $S_2$ is generally not since we do not assume that $S$ is semi-algebraic). In particular, for any $(y, x) \in T$, the following equation is satisfied $f(x, y) = y - x + \alpha g(x) = 0$. In addition, $x \in K_\alpha^c$, so that $\frac{\partial}{\partial x} f(x, y) = -I + \alpha\text{Jac } g(x)$ which is invertible since $x \in K_\alpha^c$. By the implicit function theorem, we can conclude that the fiber $T_y := \{x \mid (y, x) \in T\}$ contains isolated points. And since $T_y$ is semi-algebraic, $T_y$ is finite. We are, thus, in the position to apply Lemma A.2: There exists $N$ semi-algebraic $F_1, \ldots, F_N, N > 0$ such that:

$$T \subseteq \bigcup_{k=1}^N \text{graph } F_k.$$

In particular, this relation implies:

$$S_2 \subseteq \bigcup_{k=1}^N F_k(S).$$

The result follows by Lemma A.1. $\qquad\qquad\square$

We conclude this section with the proof of the main theorem, following standard arguments.

*Proof of Theorem 2.1.* Take $\Lambda$ as in Lemma 2.6, whose complement is finite. We prove that this $\Lambda$ is our desired set described as in Theorem 2.1. It is sufficient to prove the second condition.

Fix $\alpha \in \Lambda$ and set $\mathcal{C}_\alpha = \{x \in \mathbb{R}^n \mid G_\alpha(x) = x, \rho(\text{Jac } G_\alpha(x)) > 1\}$. For each $x \in \mathcal{C}_\alpha$, let $B_x$ be the balls corresponding to $x$ given by Theorem 2.3. In particular, we have:

$$\mathcal{C}_\alpha \subseteq \bigcup_{x \in \mathcal{C}_\alpha} B_x.$$

By Lindelof's lemma, there exists a sequence $(z_\ell)_{\ell \in \mathbb{N}}$ in $\mathcal{C}_\alpha$, such that $\mathcal{C}_\alpha \subseteq \cup_{\ell \in \mathbb{N}} B_{z_\ell}$.

Assume that the update Equation (1) initialized at a point $x_0$ and converges to a point $x \in \mathcal{C}_\alpha$. Thus, there exists natural numbers $\ell, k_0 \in \mathbb{N}$ such that for $k \geq k_0$, $G_\alpha^k(x_0) \in B_{z_\ell}$. In particular, in the light of Theorem 2.3, $G_\alpha^{k_0}(x_0) \in W_{z_\ell}^{\text{cs}}$, the local stable center manifold of $z_\ell$, given by Theorem 2.3. Since $x \in \mathcal{C}_\alpha$ was arbitrary, this shows that the set $W_\alpha$ in the statement of the theorem has the following properties

$$W_\alpha \subseteq \bigcup_{\ell \in \mathbb{N}} \bigcup_{k \in \mathbb{N}} G_\alpha^{-k}(W_{z_\ell}^{\text{cs}}).$$

For each $\ell \in \mathbb{N}$, $W_{z_\ell}^{\text{cs}}$ is a $C^1$ embedded submanifold. Using Lemma A.1, we see that $W_\alpha$ is contained in a countable union of $C^1$ embedded submanifolds as announced. $\qquad\square$

## B    PROOF OF THEOREM 2.3

The main technical tool to obtain this result is a stability theorem related to pseudo hyperbolicity Hirsch et al. (1977), which we report for Euclidean spaces. We start with the definition of pseudo-hyperbolicity.

**Definition B.1** ($\rho$-pseudo hyperbolicity). A linear map $T : \mathbb{R}^n \to \mathbb{R}^n$ is $\rho$-pseudo hyperbolic if all eigenvalues of $T$ have absolute values different from $\rho > 0$. Suppose $T$ is $\rho$-pseudo hyperbolic. In that case, we define $\mathbb{R}^n = E_{sc} \oplus E_u$ the canonical splitting of $T$ where $E_{sc}$ (resp. $E_u$) is the linear subspace induced by the eigenvectors corresponding to the eigenvalues with absolute values smaller (resp. bigger) than $\rho$.

**Theorem B.2** (Stable manifold for pseudo hyperbolic maps (Hirsch et al., 1977, Theorem 5.1)). *Consider $T :$ $\mathbb{R}^n \to \mathbb{R}^n$ a $\rho$-pseudo hyperbolic linear map and its canonical splitting $\mathbb{R}^n = E_{sc} \oplus E_u$. If $F : \mathbb{R}^n \to \mathbb{R}^n$ is a $C^1$ map, $F(0) = 0$ and the function $F - T$ has a sufficiently small Lipschitz constant $\epsilon$, then the set:*

$$W = \bigcap_{k \geq 0} F^{-k} S, \qquad S = \{(x, y) \in E_u \times E_{sc} : \|y\| \geq \|x\|\}$$

*is the graph of a $C_1$ function $g : E_{sc} \to E_u$. It is characterized by: $z \in W$ if and only if:*

$$\lim_{k \to \infty} \|F^k(z)\| / \rho^k = 0.$$

Given Theorem B.2, one can adapt localization arguments, such as (Shub et al., 1987, Chapter 5, Theorem III.7), to prove Theorem 2.3.

*Proof of Theorem 2.3.* We may assume that $p = 0$ by studying $x \mapsto F(x + p) - p$ instead of $F$. Consider a $C^\infty$ function $\varphi(x) : \mathbb{R}^n \to \mathbb{R}$ satisfying:

$$\begin{cases} \varphi(x) = 1 & \text{if } \|x\| \leq 1 \\ \varphi(x) = 0 & \text{if } \|x\| \geq 2 \end{cases}.$$

Such a function exists and it is known as a bump function. In particular, if one defines: $\varphi_s(\cdot) = \varphi(\cdot/s), s > 0$, then the function $\varphi_s$ is smooth and satisfies:

$$\begin{cases} \varphi_s(x) = 1 & \text{if } \|x\| \leq s \\ \varphi_s(x) = 0 & \text{if } \|x\| \geq 2s \end{cases}.$$

Let $T$ denote the linear mapping $x \mapsto \mathrm{Jac}\, F(0)x$, define $h = F - T$, $h_s = \varphi_s \times h$ and $F_s = T + h_s$. We have for any $s > 0$, $h_s = h$ and $F_s = F$ in the ball of radius $s$. One may choose $s > 0$ such that the Lipschitz constant of $h_s = F_s - T$ is arbitrarily small. Indeed, we have for all $x$:

$$\mathrm{Jac}\, h_s(x) = h(x)\nabla \varphi_s(x)^T + \varphi_s(x)\mathrm{Jac}\, h(x) = \frac{h(x)}{s}\nabla \varphi\left(\frac{x}{s}\right)^T + \varphi_s(x)\mathrm{Jac}\, h(x).$$

We remark that for any $s > 0$, $\mathrm{Jac}\, h_s(x) = 0$ for any $x$ such that $\|x\|_2 \geq 2s$, we will obtain a bound on a ball of radius $2s$ for $s$ small. Since $h$ is $C^1$ and $\mathrm{Jac}\, h(0) = 0$, we have:

$$\lim_{s \to 0} \sup_{\|x\| \leq 2s} \frac{\|h(x)\|}{s} = 0,$$

$$\lim_{s \to 0} \sup_{\|x\| \leq 2s} \|\mathrm{Jac}\, h(x)\| = 0.$$

Since $\varphi$ is smooth and constant outside a ball, both $\varphi_s$ and $\nabla \varphi$ are globally bounded. Therefore, we conclude that: for any $\epsilon > 0$, there exists $s > 0$ such that:

$$\|\mathrm{Jac}\, h_s(x)\| \leq \epsilon, \forall x \in \mathbb{R}^n,$$

which implies that $F_s - T$ is $\epsilon$ Lipschitz. We fix $\rho > 1$ such that the absolute values of all eigenvalues of $\mathrm{Jac}\, F(0)$ are different from $\rho$ and apply Theorem B.2 to conclude that there exists a $C^1$ function $g : E_{sc} \to E_u$ such that: $x \in \mathrm{graph}\, g$ if and only if:

$$\lim_{k \to \infty} \frac{\|F_s^k(x)\|}{\rho^k} = 0, \tag{14}$$

We define $B := B(0, s)$ and $W_{\mathrm{loc}}^{\mathrm{sc}} = \mathrm{graph}\, g \cap B$, which satisfies the requirements of Theorem 2.3. First, graph $F$ is $F_s$ invariant from the characterization Equation (14) and $F_s |_B = F |_B$ hence

$$F(\mathrm{graph}\, g \cap B) \cap B = F_s(\mathrm{graph}\, g \cap B) \cap B \subset \mathrm{graph}\, g \cap B,$$

which proves the first property. Second, if $F^k(x) \in B$ for all $k \in \mathbb{N}$, then

$$\lim_{k \to \infty} \frac{\|F_s^k(x)\|}{\rho^k} = \lim_{k \to \infty} \frac{\|F^k(x)\|}{\rho^k} = 0,$$

since $\rho > 1$ and $F^k(x)$ is bounded, that is $x \in \mathrm{graph}\, g \cap B$. The proof is concluded. $\qquad \square$

## C  DETAILS ON EIGENVALUE COMPUTATION

*Proof of Proposition 3.1.* We apply Theorem 2.1 with $D = I$ and $g = \nabla f$. Let $\lambda_1, \ldots, \lambda_m$ be the eigenvalues of $\nabla^2 f(\bar{x})$, then the eigenvalues of $D - \alpha \operatorname{Jac} g(\bar{x})$ is given by:

$$\{1 - \alpha \lambda_i \mid i = 1, \ldots, m\}$$

Therefore, $\rho(D - \alpha \operatorname{Jac} g(\bar{x})) \leq 1$ is equivalent to:

$$-1 \leq 1 - \alpha \lambda_i \leq 1, \forall i \qquad \Leftrightarrow \qquad 0 \leq \lambda_i \leq \frac{2}{\alpha}, \forall i. \qquad \square$$

*Proof of Proposition 3.2.* The proof consists of calculating the eigenvalues of the following matrix:

$$A := \begin{pmatrix} (1 + \beta)I - \alpha \nabla^2 f(x) & -\beta I \\ I & 0 \end{pmatrix}$$

We reproduce the arguments in (Polyak, 1987, Chapter 3.2).

It can be shown that for any eigenvalue $\lambda$ of $\nabla^2 f(x)$, there is a corresponding pair of eigenvalues of $A$, which are the zero of the following quadratic equation :

$$\nu^2 - \nu(1 + \beta - \alpha \lambda) + \beta = 0.$$

To make sure that all eigenvalues of $A$ has norm smaller than one, it is necessary and sufficient that:

$$0 \leq \alpha \lambda \leq 2(1 + \beta)$$

for every eigenvalue $\lambda$ of $\nabla^2 f(x)$, which proves the result. $\qquad \square$

*Proof of Proposition 3.3.* The proof of Proposition 3.3 is similar to the proof of Proposition 3.2. However, since it seems to us that the calculation of eigenvalues for the Jacobian matrix of the iteration update Equation (4) is less known, we provide a detailed derivation.

Note that $(\bar{x}, \bar{y})$ is a fixed point of Equation (4) if and only if $\bar{x} = \bar{y}$ and $\nabla f(\bar{x}) = 0$. We derive the Jacobian matrix of Equation (4) as:

$$\begin{pmatrix} (1 + \beta)(I - \alpha \nabla^2 f(\bar{x})) & -\beta(I - \alpha \nabla^2 f(\bar{x})) \\ I & 0 \end{pmatrix},$$

hence, given an eigenvalue of $\lambda$ of $\nabla^2 f(\bar{x})$, the previous Jacobian matrix possesses two corresponding eigenvalues of the following $2 \times 2$ matrix:

$$\begin{pmatrix} (1 + \beta)(1 - \alpha \lambda) & -\beta(1 - \alpha \lambda) \\ 1 & 0 \end{pmatrix}.$$

These eigenvalues are given by the roots of the following quadratic equations:

$$\nu^2 - \nu \underbrace{(1 + \beta)(1 - \alpha \lambda)}_{:= b} + \underbrace{\beta(1 - \alpha \lambda)}_{:= c} = 0.$$

Consider two possibilities concerning $\Delta := b^2 - 4c = (1 + \beta)^2 (1 - \alpha \lambda)^2 - 4\beta(1 - \alpha \lambda)$.

1. $\Delta < 0$: this condition implies that $\alpha \lambda \in [(\beta - 1)^2/(\beta + 1)^2, 1]$. When $\Delta < 0$, the quadratic equation admits two complex roots whose magnitude is given by:

$$\frac{1}{4}(b^2 + 4c - b^2) = c.$$

Hence, the condition of Theorem 2.1 reads as:

$$\beta(1 - \alpha \lambda) \leq 1 \implies \alpha \lambda \geq 1 - \frac{1}{\beta}$$

Note that when $0 < \beta < 1$, $1 - \frac{1}{\beta} < 0$. Therefore, the previous equation is satisfied automatically because $\alpha \lambda \geq (\beta - 1)^2/(\beta + 1)^2 \geq 0$.

2. Whn $\Delta \geq 0$, the quadratic equation has two real roots, given by:

$$\frac{b - \sqrt{\Delta}}{2} \qquad \text{and} \qquad \frac{b + \sqrt{\Delta}}{2}.$$

Thus, the condition of Theorem 2.1 reads as:

$$\frac{b - \sqrt{\Delta}}{2} \geq -1,$$

$$\frac{b + \sqrt{\Delta}}{2} \leq 1.$$

Solving both inequalities:

$$\frac{b - \sqrt{\Delta}}{2} \geq -1 \implies b + 2 \geq \sqrt{\Delta}$$
$$\implies b^2 + 4b + 4 \geq \Delta = b^2 - 4c$$
$$\implies (1 + \beta)(1 - \alpha\lambda) + 1 \geq -\beta(1 - \alpha\lambda)$$
$$\implies (1 + 2\beta)(1 - \alpha\lambda) + 1 \geq 0$$
$$\implies \frac{2 + 2\beta}{1 + 2\beta} \geq \alpha\lambda.$$

$$\frac{b + \sqrt{\Delta}}{2} \leq 1 \implies 2 - b \geq \sqrt{\Delta}$$
$$\implies b^2 - 4b + 4 \geq \Delta = b^2 - 4c$$
$$\implies (1 + \beta)(1 - \alpha\lambda) - 1 \leq \beta(1 - \alpha\lambda)$$
$$\implies (1 - \alpha\lambda) - 1 \leq 0$$
$$\implies \alpha\lambda \geq 0.$$

By combining all these observations, we conclude that $0 \leq \alpha\lambda \leq \frac{2+2\beta}{1+2\beta}$, which yields the result. $\square$

*Proof of Proposition 3.4.* We only consider the case $\beta = 0$ (cf. Equation (5)). The computation similarly extends to general $\beta$ as in Proposition 3.2.

We compute the Jacobian matrix the update rule Equation (5):

$$I - \alpha\nabla^2 f(\bar{x} + \rho \underbrace{\nabla f(\bar{x})}_{=0})(1 + \rho\nabla^2 f(\bar{x})) = I - \alpha\nabla^2 f(\bar{x})(1 + \rho\nabla^2 f(\bar{x}))$$

and their eigenvalues, given by:

$$1 - \alpha\lambda(1 + \rho\lambda),$$

where $\lambda$ is an eigenvalue of $\nabla^2 f(\bar{x})$. Applying Theorem 2.1 simply yields the result. $\square$

*Proof of Proposition 3.5.* Similar to the proof of Proposition 3.4, we only consider the case $\beta = 0$ (cf. Equation (9)) and the computation for general $\beta$ (cf. Equation (10)) can be done similarly as in Proposition 3.2. Consider $\bar{x}$ a critical point of $f$. Computing the Jacobian matrix of the iteration update of Equation (9) yields:

$$H := I - \alpha\nabla^2 f(\bar{x})(1 + \rho\nabla^2 f(\bar{x}))^2.$$

Note that for an eigenvalue of $\lambda$ of $\nabla^2 f(\bar{x})$, $H$ has an eigenvalue equal to $1 - \alpha\lambda(1 + \rho\lambda)^2$. Applying Proposition 3.4 yields the result immediately. $\square$

*Proof of Proposition 3.7.* Similar to Proposition 3.5. $\square$

## D    APPEARANCE OF NEW FIXED POINTS

For $K > 0$, set

$$f(x) = \begin{cases} 0 & x \leq 1 \\ -\dfrac{K}{3}(x-1)^3 & x > 1, \end{cases}$$

so that $\text{crit} f = (-\infty, 1]$ where all points are local minimizers except $x = 1$. One has $f'(x) = 0$ for $x \leq 1$, while $f'(x) = -K(x-1)^2$ for $x > 1$. To model USAM, set $T(x) := x + \rho f'(x)$, so that

$$T(x) = \begin{cases} x & x \leq 1, \\ x - \rho K(x-1)^2 & x > 1. \end{cases}$$

Observe that $x \in T^{-1}(\mathrm{crit} f)$ iff $T(x) \leq 1$.

For $x > 1$, write $s := x - 1 > 0$; the condition becomes

$$1 + s - \rho K s^2 \leq 1 \iff s(1 - \rho K s) \leq 0 \iff s \geq \frac{1}{\rho K}.$$

Hence $T^{-1}(\mathrm{crit} f) \cap (1, +\infty) = \left[ 1 + \frac{1}{\rho K}, +\infty \right)$. This shows that USAM creates an infinite length interval of fixed points.

