# OpenReview forum: "CONVERGENCE OF OPTIMIZERS IMPLIES EIGENVALUES FILTERING AT EQUILIBRIUM"
_ICLR.cc/2026/Conference — Submitted to ICLR 2026_

### Official Review · Reviewer_ocNd · 2025-10-27

**Soundness:** 3
**Presentation:** 3
**Contribution:** 1
**Rating:** 2
**Confidence:** 5

**Summary:**

In this work the authors develop a general framework for analyzing the properties of the loss landscape after convergence. Their methodology uses tools from analysis/dynamical systems theory to derive the geometric properties of any converged points. Their framework recapitulates known results for GD and some momentum variants, as well as some SAM variants.

The authors propose some SAM variants with different properties at convergence according to their methods, and provide some experimental evidence that these variants have stronger regularization properties.

**Strengths:**

The paper provides a very clean general setup for analyzing the loss landscape properties at convergence. In addition all the main theoretical results are cleanly presented and generally reference the previous works well.

**Weaknesses:**

The main weakness of the work is that it doesn't give any actionable lessons to the community working on sharpness regularization. The method called Hessian SAM in the paper is known in the literature (e.g. as Penalty SAM here [1]). The 2-step SAM has no benefits for a large computational cost. The general idea that SAM modifies the edge of stability (and therefore converged eigenvalues) is already known. While having a more precise theoretical characterization is nice, many of these results come quite naturally from Taylor expansion of losses around stationary points.

The experiments use minibatching, which can strongly affect the curvature dynamics, particularly as it can become unclear if networks actually reach convergence in this regime. In addition, some of the experimental settings use ReLU, which has to be treated very carefully with what the authors call Hessian SAM (Penalty SAM in the reference) [1].



[1] https://proceedings.neurips.cc/paper_files/paper/2024/hash/ee3ce0121939f42098cdefd3ea025bf1-Abstract-Conference.html

**Questions:**

How does proposition 3.7 compare to the edge of stability bound in [1], e.g. Equation 16? there appears to be a potential discrepancy of a factor of $\lambda$ in the term linear in $\rho$.

Hessian USAM has been studied elsewhere, particularly in [2] where it is called "Penalty SAM". How do the results of [2] relate to the results in the paper?

[1] https://proceedings.mlr.press/v202/agarwala23a

[2] https://proceedings.neurips.cc/paper_files/paper/2024/hash/ee3ce0121939f42098cdefd3ea025bf1-Abstract-Conference.html

---

> ### Author Response · Authors · 2025-11-18
>
> We thank the reviewer for their reading and acerbic comments. It will definitely help us improve the paper, in particular by showing that we need to provide more detail on the interaction between algorithmic linearization and local behaviour, so as to avoid confusion and false simplicity.
>
> **Weaknesses comments:**
> 1. ``*The main weakness of the work is that it doesn't give any actionable lessons to the community working on sharpness regularization.*''
>
> We are extremely puzzled by the remark that the work offers no actionable lessons. Our main theorem gives an explicit spectral-radius inequality, which directly quantifies the level of eigenvalue filtering induced by the algorithm and thus the sharpness it induces. This inequality translates into concrete bounds on admissible curvature as a function of the hyperparameters and, as our experiments illustrate, accurately predicts the strength of filtering observed in practice. Here we must warn against the idea that one “would just need to apply Taylor’s formula to obtain these results,” an idea that cannot be invoked since this approach *abounds with counterexamples* in dynamical systems, even under uniqueness and hyperbolicity of critical points (see below). Our paper precisely justifies when and why Taylor’s formula is actually actionable.* Tables provided in our preamble show how useful a rigorous theorem is for deriving a wealth of new results.
>
> A practical instance of an actionable implication of our result is, for example, that: given two optimization algorithms with their fixed hyper-parameters, one can infer which one of algorithm is likely to find solutions with better sharpness (assuming that these two converges), just by applying our theoretical results and doing the computations as we did in the paper. That is why we proposed two algorithms in the last section: although they are more expensive to run (as pointed out by the reviewer), our goal is simply to test our theoretical results with newer algorithms, and not just famous algorithms such as GD or heavy ball.
>
> 2. ``*The method called Hessian SAM in the paper is known in the literature (e.g. as Penalty SAM here [1]). The 2-step SAM has no benefits for a large computational cost. The general idea that SAM modifies the edge of stability (and therefore converged eigenvalues) is already known. While having a more precise theoretical characterization is nice, many of these results come quite naturally from Taylor expansion of losses around stationary points.*''
>
> --- We discuss SAM methods (SAM, Penalty SAM and Hessian SAM, 3 different methods) in the next comment.
>
> --- As commented by the reviewer, the idea of SAM modifying the Edge-of-stability is, indeed, not new in the literature (e.g., [1,2]). However, as we explicitly explain in our preliminary comment, our theorem does not just provide the same conclusion with nicer assumptions, but it does show a much stronger property (non-convergence to certain optimal solutions), with a very general setting. For a concrete example, let's consider [1, Theorem 2.2]: they showed that if $f$ is the square loss of an overparameterized regression models, then there exists an initialization $x_0$ close to an optimal solution $x^\star$ such that the iterates initialized by $x_0$ will eventually be driven away from $x^\star$. This property is weaker than ours (non-convergence to $x^\star$ with probability $1$).
>
> --- We must disagree with the suggestion that unjustified Taylor expansions suffice to obtain our results. Using Taylor expansions without proper regularity assumptions would, for instance, trivialize the proofs of the inverse and implicit function theorems, of the Morse lemma, and simplify considerably the proof of Lojasiewicz inequalities. We just quote a few cases, there are many more in different fields. Even worst, it would also lead to incorrect conclusions in the very field underlying our work! For example, by wrongly asserting that a nonlinear flow and its linearized flow are equivalent (even in the polynomial case). This can be seen from Poincaré’s resonance conditions for linearizing analytic fields. As explained before, we precisely justify the use of Taylor expansions (we could even say, that in some sense, it is the object of this paper). We hope this discussion will be carefully taken into account.

---

> ### Author Response · Authors · 2025-11-18
>
> **Weakness comments** (continue)
>
> 3. ``*The experiments use minibatching, which can strongly affect the curvature dynamics, particularly as it can become unclear if networks actually reach convergence in this regime. In addition, some of the experimental settings use ReLU, which has to be treated very carefully with what the authors call Hessian SAM (Penalty SAM in the reference) [1].*"
>
> We definitely agree with the reviewer that our experimental setting (minibatch GD, with non-smooth component) is different from our theoretical results (GD with smooth function). Nevertheless, the sharpness we obtained are still well-predicted by our theoretical results. We will provide more detailed discussion concerning this aspect in the next version of the paper.
>
> **Questions:**
> 1. ``*How does proposition 3.7 compare to the edge of stability bound in [1], e.g. Equation 16? there appears to be a potential discrepancy of a factor of  in the term linear in $\rho$*"
>
> In [1], the bound for SAM is $\lambda (1 + \rho\lambda) \leq \frac{2}{\alpha}$ while Proposition 3.7 applied to SAM (without momentum) is:
> $$\lambda (1 + \rho \lambda^2) \leq \frac{2}{\alpha}$$
> where $\alpha$ is the step-size, $\lambda$ is  any eigenvalue of the Hessian and $\rho$ is SAM inner step-size. The discrepancy is to be expected since the Hessian SAM has a different asymptotic behavior in comparison to SAM.
>
> 2. ``*Hessian USAM has been studied elsewhere, particularly in [2] where it is called "Penalty SAM". How do the results of [2] relate to the results in the paper?*""
>
> In fact, Penalty SAM [2] and Hessian SAM are different. Indeed, the formula for Penalty SAM is:
> $$x_{k+1} = x_k - \alpha \left(\nabla f(x_k) + \frac{1}{\|\nabla f(x_k)\|}\nabla^2 f(x_k) \nabla f(x_k)\right)$$
> while that of Hessian SAM is given by:
> $$x_{k+1} = x_k - \alpha \nabla f(x_k + \rho \nabla^2 f(x_k) \nabla f(x_k))$$
> The Hessian term is inside the gradient evaluation, which is different from Penalty SAM. We do not find any other link between those two algorithms apart from the obvious step discrepancy.
>
> [1] Atish Agarwala and Yann Dauphin. Sam operates far from home: eigenvalue regularization as a dynami- cal phenomenon. In Proceedings of the 40th International Conference on Machine Learning, ICML’23.
>
> [2] Yann N. Dauphin, Atish Agarwala, Hossein Mobahi, Neglected Hessian component explains mysteries in sharpness regularization. Advances in Neural Information Processing Systems 37 (NeurIPS 2024)
>
> [3] Kwangjun Ahn, Jingzhao Zhang, and Suvrit Sra. Understanding the unstable convergence of gradient descent. In International Conference on Machine Learning, pp. 247–257. PMLR, 2022.
>
> [4] Zhanpeng Zhou, Mingze Wang, Yuchen Mao, Bingrui Li, and Junchi Yan. Sharpness-aware minimiza- tion efficiently selects flatter minima late in training. In Proceedings of the International Confer- ence on Learning Representations, ICLR, January 2025.

---

> > ### Comment · Reviewer_ocNd · 2025-11-20
> > **Response to authors**
> >
> > I thank the authors for their response to all reviewers; I will respond here.
> >
> > Thank you for the clarification on Hessian USAM. I had missed where the Hessian-gradient product was applied. No further questions on that front.
> >
> > I appreciate the authors' argument about the value of their work; indeed convergence can be a tricky subject in non-linear, discrete dynamical systems. I do not contest that the theorems are more detailed than simply saying "just Taylor expand".
> >
> > That being said: I stand by my previous statement that the utility of the work in its current form to the area of sharpness regularization research seems low. The authors claim that use of Taylor expansions was previously unjustified, except in particular cases which were proven ad-hoc, and that their analysis unifies and expands the regimes where this method rigorously works. I don't dispute this, but wanted to point out that the theoretical results themselves don't actually cover the primary training scenarios, which have properties including:
> >
> > * Non semi-algebraic functions (or indeed, non-differentiable functions).
> > * Lack of convergence during training
> > * Various sources of stochasticity (most notably mini-batching)
> >
> > In this sense, the theorems, though rigorous, are technically "unjustified" for application to ~ any real world training scenario.
> >
> > For now I will keep my score; however I am carefully monitoring the discussion with the other reviewers and am open to changing my mind depending on how those discussions go.

---

> ### Author Response · Authors · 2025-11-21
>
> We thank the referee for her/his answer, we are very grateful of having the possibility to exchange, it is extremely precious. We would like to make precise a few things and to raise perhaps some questions.
>
> >  I don't dispute this, but wanted to point out that the theoretical results themselves don't actually cover the primary training scenarios, which have properties including:
> > Non semi-algebraic functions (or indeed, non-differentiable functions).
>
> The definability aspect brought up in our first answer is crucial. We are not aware of any (smooth or nonsmooth) deep-learning problem, restricted to compact sets, whose ingredients are not definable. In particular, all real-analytic functions are covered, but also essentially all standard pipelines built from common activation functions and losses: ReLU, leaky-ReLU, PReLU, ELU, SELU, GELU, softplus, softsign, sigmoid, tanh, hard-sigmoid, hard-tanh, Swish, Mish, maxout, etc., as well as usual losses (squared loss, cross-entropy, logistic loss, hinge-type losses) and their compositions with linear maps, convolutions, pooling, residual connections, transformers... All these constructions are definable on compact subsets, and thus are encompassed by our framework (see the discussion and references in our general answer). Therefore we do not see what we would gain by going beyond this.
>
> As for nonsmooth extensions of our results we are unsure of what should be done according to the referee. Our current theorems are formulated in terms of gradients and Hessians, so extending them to a nonsmooth setting is really nontrivial. In particular, sharpness can become infinite, gradients are no longer Lipschitz … Designing an analogue of our spectral-radius theorem appears very difficult to formulate ... we believe it deserves a separate research program.
>
> > Lack of convergence during training
>
> Here we simply recall the following reformulation of our main result as given in the discussion with UZZ3:
>
> Theorem 1.1. Let $D,g$ satisfy the conditions of Theorem 1.1. For generic step size $\alpha>0$ and random initialization, the probability of converging to a point $\bar x$  such that  $\rho(\mbox{Jac}_x ( D-\alpha g )( \bar x))>1$  is zero, where $\rho$ denotes the spectral radius.
>
> Thus, there is an equivalent reformulation without the convergence assumption that bothers the referee. We are grateful to the referees for allowing us to bring up all the facets of our main result. Of course, as explained to referee UZZ3, we will include this in our revision.
> > Various sources of stochasticity (most notably mini-batching)
>
> We agree that in practice there are many sources of stochasticity as mini-batching. However, even for deterministic algorithms, the general theory of dynamical systems is already very subtle, and controlling the dependence on stepwise choices is extremely challenging (see below also). We therefore view the passage from “linear heuristics” to a genuinely nonlinear, fully deterministic dynamical-system analysis as a mathematical achievement in itself.  Not to speak of the fact that the very theory of linearization of stochastic method is far from complete.
> > In this sense, the theorems, though rigorous, are technically "unjustified" for application to ~ any real world training scenario.
>
> As a preamble, let us provide a simple example of an optimization dynamics that does not lend itself to linearization:
> $$x'(t) = -x(t), \qquad y'(t) = -2y(t) + a x(t)^2.$$
> This is the perturbed system of the gradient system of the function $f(x,y)= \tfrac12 x^2 + y^2$:
> $$x'(t) = -x(t),  y'(t) = -2y(t).$$
> This system and its linearization allow us to minimize $f(x,y)$.
> The solutions of the perturbed system are
> $$
> x(t) = x_0 e^{-t},
> y(t) = e^{-2t}\bigl(y_0 + a t x_0^2\bigr).
> $$
> Despite the fact that the ODE has a unique equilibrium and a diagonal Jacobian with eigenvalues -1 and -2, the convergence velocity is not of the same order as in the linear system: the resonant term $a x^2$ produces the additional factor $at$. Therefore, the convergence rate of $y$ to $0$ is $O(t e^{-2t})$, instead of $O(e^{-2t})$ as in the classical gradient system.
>
> For general systems, the situation becomes rapidly out of control. Therefore linearization is not a trivial subject as it can change *convergence rates* (for instance).
>
> Secondly, let us point out that most results in the sharpness field are quadratic for $f$ (see tables), meaning they are linear in terms of ODES. Therefore the delicate problem of nonlinearities is simply avoided!
>
> With this in mind, we would like to emphasize that our paper is mathematical and its intent is to provide solid grounds for a good way of  linearizing algorithms in training. Our choice to submit to ICLR is motivated by this necessity, by the belief it is of interest to other scientists, and by the fact that linearization is delicate, and sometimes misleading, as the example above illustrates. We will make this point clearer in the revision

---

### Official Review · Reviewer_1fUe · 2025-10-28

**Soundness:** 2
**Presentation:** 2
**Contribution:** 2
**Rating:** 4
**Confidence:** 4

**Summary:**

The paper proposes a unified theoretical perspective to analyze the convergence of various optimization algorithms. Building on this perspective, the authors introduce two new optimizers, USAM2 and HUSAM, and provide convergence results. Experimental validation is conducted on MNIST and CIFAR10 using standard architectures.

**Strengths:**

1. The idea of viewing the convergence of different optimizers through a unified lens is interesting and conceptually valuable.

2. The theoretical development for the proposed USAM2 and Hessian-USAM algorithms is well-presented.

**Weaknesses:**

1. Aside from the authors’ own methods (USAM2 and HUSAM), the theoretical results for other optimizers seem largely covered by prior work, so the novelty is limited.

2. Experimental evaluation is weak. MNIST is too small to provide convincing evidence, and the performance gains on MNIST of USAM2 and HUSAM, especially in accuracy, are very marginal.

3. The WideResNet-16-8 on CIFAR10 setting is outdated. In this setting, USAM2 even underperforms USAM. Although HSAM shows a more noticeable improvement, it requires second-order Hessian information. It is questionable whether such a method is practical or valuable for large-scale neural networks.

4. The empirical section lacks experiments on modern benchmarks and larger-scale models where the claimed contributions would matter.

**Questions:**

See Weakness.

---

> ### Author Response · Authors · 2025-11-18
>
> We thank the reviewer for her/his comments, they will definitely help us improve the paper. We agree that our presentation needs refinement and additional precision to convey the strengths and novelties of our contributions.
>
> **Weaknesses comments:**
> 1. ``*Aside from the authors’ own methods (USAM2 and HUSAM), the theoretical results for other optimizers seem largely covered by prior work, so the novelty is limited.*"
>
> We need to disagree, as shown in our preliminary table: the improvements are considerable, and in some cases our results are entirely novel. Moreover, the principles we establish open the door to the analysis of virtually any new local (e.g., first/second-order ...) algorithm. This was what guided our introduction of SAM variants. We also need to mention that, although the results may appear natural once the path is laid out, that the proofs are non trivial as often for dynamical systems.
>
> As mentioned to referee UZZ3, our framework also offers new insights into EoS (see our response). In particular, our bounds on the Hessian spectrum for classical methods match those in the foundational EoS work, and when the EoS conjecture is interpreted as a saturation of the spectral radius as a function of the step size, our results rigorously establish one of the two key inequalities. This connection will be made more explicit in the revised version.
>
>
> 2. ``*Experimental evaluation is weak. MNIST is too small to provide convincing evidence, and the performance gains on MNIST of USAM2 and HUSAM, especially in accuracy, are very marginal.*''
>
> 3. ``*The WideResNet-16-8 on CIFAR10 setting is outdated. In this setting, USAM2 even underperforms USAM. Although HSAM shows a more noticeable improvement, it requires second-order Hessian information. It is questionable whether such a method is practical or valuable for large-scale neural networks.*''
>
> 4. ``*The empirical section lacks experiments on modern benchmarks and larger-scale models where the claimed contributions would matter.*''
>
> Answers to 2/3/4. We thank the reviewer for these comments: we did not explain well why we proposed two new algorithms and conducted several experiments. We shall improve this aspect in our revised version. We want to empirically test the following: given two optimization algorithms with fixed hyper-parameters, can one infer which one is likely to find solutions with smaller sharpness (assuming that they converge), just by applying our theoretical results and doing the computations as we did in the paper. That is the reason why we proposed two algorithms in the last section: although they are much more expensive to run, our theory predicts that it has a better effect at improving the sharpness. The experiments only illustrate this empirically.
>
> Therefore, while the models and the experiments might fail modern practical Deep Learning standards, we find that our experiments still reaches the goals. We hope that our article is mainly judged as a theoretical one, since our arguments and experiments were developped towards this direction.

---

> > ### Comment · Reviewer_1fUe · 2025-11-26
> >
> > Thank you for your clarifications. However, as an ICLR submission, it is typically expected that the theoretical contributions demonstrate clear relevance and practical significance within modern neural networks.
> >
> > As you also emphasize that your work should be “mainly judged as a theoretical one,” it may be more suitable for venues that focus primarily on optimization theory rather than machine learning applications.
> > In particular, since your contributions seem largely independent of neural networks and do not require deep-learning-specific structures, their theoretical value would likely be more appropriately assessed by the optimization theory community. Journals or conferences such as COLT, Operations Research, or Mathematical Programming may therefore provide a better fit and a more receptive audience for the type of contributions you aim to make.
> >
> > Currently I would keep my score but reduce my confidence to 3.

---

> > > ### Author Response · Authors · 2025-11-27
> > >
> > > The referee claims that "as an ICLR submission, it is typically expected that the theoretical contributions demonstrate clear relevance and practical significance within modern neural networks.". This argument is not in line with the [call for papers](https://iclr.cc/Conferences/2026/CallForPapers). It is also not in line with the [ICLR 2025 topics](https://iclr.cc/virtual/2025/papers.html?filter=topic&search=Theory-%3EOptimization) which includes “Theory → Optimization”.
> > >
> > > This being recalled, we provide unified and simple guarantees for sharpness estimation for a broad class of algorithms. This is clearly relevant to the practice of modern neural networks [1,2] and significantly improves upon existing theoretical results published in ML venues. The paper therefore does meet the referee’s stated requirements.
> > >
> > > At this stage of the discussion, the main points of the original evaluation of the referee have been addressed and the score is no longer based on scientific arguments. We believe that the paper deserves a reassessment—possibly a negative one—but based on genuine arguments, in accordance with ICLR policy.
> > >
> > > [1] Jeremy Cohen, Simran Kaur, Yuanzhi Li, Zico Kolter, and Ameet Talwalkar. Gradient descent on neural networks typically occurs at the edge of stability. ICLR 2021.
> > > [2] Zhanpeng Zhou, Mingze Wang, Yuchen Mao, Bingrui Li, and Junchi Yan. Sharpness-aware minimization efficiently selects flatter minima late in training. ICLR 2025.

---

### Official Review · Reviewer_UZZ3 · 2025-10-31

**Soundness:** 3
**Presentation:** 2
**Contribution:** 3
**Rating:** 4
**Confidence:** 4

**Summary:**

This paper provides an interesting new perspective on the properties that the convergence point should satisfy under the assumption that the algorithm converges. The authors present a general statement based on the Stable Manifold Theorem, showing that the corresponding spectral radius is at most 1. They further apply their theory to several different algorithms, including gd, usam, and momentum variants, and demonstrate that different algorithms essentially select minima with different degrees of flatness.

**Strengths:**

This paper establishes an interesting and novel theoretical framework that studies a “dual” version of stability analysis, in which the algorithm is assumed to converge and the focus is on the properties of the initialization and the solution. The theory is built upon a generalized Hadamard–Perron stable manifold theorem. In contrast to stability analyses that require local diffeomorphism assumptions, this work relies on much milder conditions, such as the requirement of only semi-algebraic functions, and remains valid for large step sizes. The authors apply their theory to various practical algorithms, demonstrating different eigenvalue filtering effects and providing experimental results that validate their theoretical claims. Overall, the arguments presented in this paper are sound and well supported.

**Weaknesses:**

Although I enjoyed reading Section 2 of this paper, there are several shortcomings. The authors repeatedly claim that their theory is built upon weaker assumptions; however, their framework requires an additional assumption of algorithmic convergence, which is itself non-trivial and implicitly imposes constraints. For example, while Theorem 2.1 is stated to hold for large (even unbounded) step sizes, excessively large step sizes would clearly lead to divergence.

More importantly, to the best of my knowledge, the analyses for different algorithms in this paper do not appear to yield new results. For instance, Proposition 3.1 does not differ in essence from the well-known stability condition, and Proposition 3.4 does not go beyond the findings of Zhou et al. (2025), who analyzed the stability of USAM in detail. Although the theoretical perspective is interesting, it seems to offer limited new insights to the machine learning community.

**Questions:**

The class of semi-algebraic functions does not include common deep learning activation functions such as sigmoid and tanh. Although the authors claim that their results can be extended to all smooth losses, it remains unclear why this was not done, which may give the impression that the work is somewhat incomplete.

The authors mention the connection to EOS several times, but the discussion is too brief. Would it be possible to conduct a deeper analysis to provide new insights, or include some related experimental evidence?

Since the results presented in this paper are for the full-batch setting, why are stochastic variants still used in the experiments?

Minor: The definition on Semi-algebraic sets and functions should be moved to the beginning of Section 2 to aid understanding.

---

> ### Author Response · Authors · 2025-11-18
>
> **General comment:**
>
> We thank the reviewer for his/her constructive critics that we address below. They will definitely help us to improve the presentation of the paper.
>
> **Weaknesses comments:**
> 1. ``*The authors repeatedly claim that their theory is built upon weaker assumptions; however, their framework requires an additional assumption of algorithmic convergence, which is itself non-trivial and implicitly imposes constraints. For example, while Theorem 2.1 is stated to hold for large (even unbounded) step sizes, excessively large step sizes would clearly lead to divergence.*"
>
> Many thanks, this remark is extremely useful as it reveals a serious misunderstanding we need to address! If we adopt  the standard statements used by the "sharpness community" Theorem 1.1 can be reformulated as:
>
> *Let $D,g$ satisfy the conditions of Theorem 1.1. For generic step size $\alpha > 0$ and random initialization, the probability of converging to a point $x^\star$ such that $\texttt{Jac}_x(D - \alpha g)(x^\star) > 1$ is zero*
>
> Most results we cite are formulated in this way, which indeed `removes' the convergence assumption, explaining the confusion.  Please observe that it is actually uniquely a matter of presentation. In our article, because we aim at  highlighting the eigenvalue-filtering mechanism of optimization algorithms, we chose the formulation currently used, but we shall provide both in our revision.
>
> 2. ``*More importantly, to the best of my knowledge, the analyses for different algorithms in this paper do not appear to yield new results. For instance, Proposition 3.1 does not differ in essence from the well-known stability condition, and Proposition 3.4 does not go beyond the findings of Zhou et al. (2025), who analyzed the stability of USAM in detail. Although the theoretical perspective is interesting, it seems to offer limited new insights to the machine learning community.*''
>
>  We kindly disagree with the first part of the comment, as we think provide genuinely new results. The novelties are emphasized in the tables of our preliminary statements, we hope they speak for themselves. The paper will be revised accordingly. These tables show that many new results are brought up, with a systematic approach to the subject through the spectral radius of nonlinear algorithms.
>  Concerning the results in [1], they are quite different: while ours shows that deterministic SAM avoids optimal solutions whose eigenvalues do not satisfy certain inequality for a wide class of functions, the linear-stability results in [1] are focused on stochastic SAM applied to linear regression [1, equation (5) ,Theorem 4.1, Corollary 4.2, Section C.1]. We find that the two results are not comparable since linear stability is defined via a first-order Taylor linearization of the model [1, Equation 5], effectively resulting in a linear regression problem. The paper contains no argument regarding whether or not non-convergence is implied by linear stability and vice versa.
>
> [1] Zhanpeng Zhou, Mingze Wang, Yuchen Mao, Bingrui Li, and Junchi Yan. Sharpness-aware minimiza- tion efficiently selects flatter minima late in training. In Proceedings of the International Conference on Learning Representations, ICLR, January 2025.

---

> ### Author Response · Authors · 2025-11-18
>
> **Questions:**
>
> 1. ``*The class of semi-algebraic functions does not include common deep learning activation functions such as sigmoid and tanh. Although the authors claim that their results can be extended to all smooth losses, it remains unclear why this was not done, which may give the impression that the work is somewhat incomplete.*''
>
>
> We thank the reviewer for this remark, we need indeed to improve that aspect. We formulated Theorem 1.1 under semi-algebraicity mainly for pedagogical reasons, as it avoids introducing the broader and more technical framework of definable functions. This choice was purely one of convenience: the result actually holds in full generality for definable mappings, a class that includes essentially all activation functions used in practice—such as sigmoid, tanh, softplus, GELU, and many others. In the revised version, we will state the theorem directly in this more general and powerful definable setting. We pointed references illustrating this aspect more explicitly in our preliminary statement.
>
>
> 2. ``*The authors mention the connection to EOS several times, but the discussion is too brief. Would it be possible to conduct a deeper analysis to provide new insights, or include some related experimental evidence?*""
>
> The referee is absolutely right and we will add a paragraph on that question. Neural network training at EoS is the conjugacy of an overall loss decrease, together with the saturation of the stepsize at its maximal possible level (given by second order information) that we could summarize in step=2/sharpness. We rigorously showed that this equation is exactly spectral radius of algorithm=1 and that convergence implies step $\leq$ 2/sharpness with probability one. The connection with usual Edge-of-Stability (EoS) results goes then through all our upper bounds on the Hessian spectrum for classical algorithms —gradient descent, heavy ball, and Nesterov acceleration—match those in the original EoS paper [2].
>
> To deepen the connection between our work and EoS, we shall also describe and reformulate the saturation mechanism for the sharpness aware algorithms in our spectral radius context, as in current experiments (showing saturation in experiments). These will be reported in a new subsction in the revision of our paper.
>
> 3. ``Since the results presented in this paper are for the full-batch setting, why are stochastic variants still used in the experiments?"
>
> Our point was simply to propose experimental runs with standard procedures while economizing compute, but we can certainly revert to the full-batch approach.
>
> [2] Jeremy Cohen, Simran Kaur, Yuanzhi Li, J Zico Kolter, and Ameet Talwalkar. Gradient descent on neural networks typically occurs at the edge of stability. In International Conference on Learning Representations, 2021.

---

> > ### Comment · Reviewer_UZZ3 · 2025-11-27
> >
> > Thank you for the authors’ response, which has resolved most of my concerns. However, as the authors themselves acknowledge, the current version of the manuscript requires substantial revisions before it can reach a level suitable for publication. For the sake of fairness, I will keep my original score. I would not object to the paper being accepted by the conference, but even if it is rejected, I encourage the authors to submit a revised version to future machine learning venues, as I believe the paper offers an interesting new perspective.

---

> > > ### Author Response · Authors · 2025-11-27
> > >
> > > We  thank the referee but we strongly disagree with her/his statements: there are no substantial revision to make. The reformulation of our theorem is immediate, the addition of a few examples on definability is routine (see refs), the proofs are *unchanged* as definability is a qualitative generalization of semi-algebraicity, relation to EoS is also trivial since the inequality on the radial spectrum is the inequality of EoS. Overall, the main subject of our revision is to implement more explicit discussions in the paper to avoid misunderstanding and emphasize the novelty and generality of our results (e.g., as we already did in our general comments). It takes one hour of two of work.
> > >
> > > The referee acknowledges having missed the theoretical part: she/he believes that we use additional assumptions, which is not the case (we actually use fewer). She/he also states that “the analyses for different algorithms in this paper do not appear to yield new results,” which is factually incorrect, as clearly shown in the tables. Since these are the main criticisms, out of fairness, we believe that the paper deserves a reassessment based on genuine arguments, in accordance with ICLR policy and with the present thread, starting from our first answer.

---

### Official Review · Reviewer_crYV · 2025-11-01

**Soundness:** 3
**Presentation:** 3
**Contribution:** 2
**Rating:** 6
**Confidence:** 3

**Summary:**

The paper investigates the following optimization question: assuming an algorithm converges, what properties does its limiting point satisfy? The authors show that at any convergence point, the Jacobian of the update map has a spectral radius $\le 1$, which imposes eigenvalue filters on the Hessian that depend on the hyperparameters. They then apply this general result to several examples, including Gradient Descent, Heavy Ball, Nesterov’s method, and Unnormalized Sharpness-Aware Minimization (USAM). For USAM, the filter can admit negative eigenvalues, suggesting possible convergence to saddle points. Motivated by this observation, the authors propose Two-step USAM and Hessian USAM, whose filters exclude negative curvature, thus avoiding strict saddles. Small-scale experiments (MLPs on MNIST/Fashion-MNIST and WRN on CIFAR-10) show smaller top Hessian eigenvalues at convergence and similar performance.

**Strengths:**

-	Understanding the implicit bias or structure of the minima that optimization algorithms converge to is an interesting research problem.
-	The paper’s main contribution is to characterize the geometry of the minima to which the algorithm can converge (assuming convergence). The assumption is quite general and covers many relevant settings.
-	Overall, the paper is well written and easy to follow.

**Weaknesses:**

-	The proposed algorithms, Two-step USAM and Hessian USAM, appear to require more computation or backpropagation per iteration than SAM. Since SAM updates are already relatively expensive, the proposed methods may be slower for large-scale applications.
-	The experiments conducted on small-scale datasets are not sufficient to support the empirical claims, especially it is not clear to see the difference between the performance. Larger-scale experiments would better validate the results.

**Questions:**

-	Is there any analysis or discussion of the additional complexity of the proposed algorithms compared to SAM?
-	Can the results be extended to the stochastic setting or to adaptive methods such as Adam?
-	If additional structure of the problem is known (e.g., certain smoothness assumptions on the objective function), could the results be strengthened to hold for all parameters instead of almost surely?
-	Is there a robustness version of the results? For $x$ near the limiting point, is the Jacobian’s spectral radius bounded by $1+\delta$ for some small $\delta$?
-	It would be helpful to include more details about the experimental setup in the appendix. For example, how is the performance of different algorithms compared fairly. Are they run for the same number of epochs or the same number of backpropagations (since the proposed algorithm seems to require more backpropagations per iteration)?

---

> ### Author Response · Authors · 2025-11-18
>
> We greatly thank the reviewer for his/her constructive feedbacks.
>
> **Weakness comments:**
> ``*The proposed algorithms, Two-step USAM and Hessian USAM, appear to require more computation or backpropagation per iteration than SAM. Since SAM updates are already relatively expensive, the proposed methods may be slower for large-scale applications.
> The experiments conducted on small-scale datasets are not sufficient to support the empirical claims, especially it is not clear to see the difference between the performance. Larger-scale experiments would better validate the results.*''
>
> We thank the reviewer for these comments: our experiments are merely illustrative, we did not explain this well enough. We want to empirically test the following: given two optimization algorithms with fixed hyper-parameters, can one infer which one is likely to find solutions with smaller sharpness (assuming that they converge), just by applying our theoretical results and doing the computations as we did in the paper. That is the reason why we proposed two algorithms in the last section: although they are much more expensive to run, our theory predicts that it they have a better effect at improving the sharpness, which we illustrate in the experiments.
>
> **Questions:**
> 1. "*Is there any analysis or discussion of the additional complexity of the proposed algorithms compared to SAM?*"
>
> The complexity of these two new algorithms are the same as the gradient descent up multiplicative constant. We will make precise these estimations in the next version of our paper.
>
> 2. "*Can the results be extended to the stochastic setting or to adaptive methods such as Adam?*""
>
> There has been several attempts at describing the edge of stability phenomenon for stochastic algorithms [1,2] or adaptive methods [3]. These have been described empirically or for specific settings (see also the table above). Obtaining a general result, similar to ours, covering adaptive and stochastic methods would certainly build on these works. However, this would not be direct at all, requiring many additional developments. This constitute a relevant extension of our work and we are unsure about how far this could be pushed. We will mention this in the conclusion.
>
>
> 3. "*If additional structure of the problem is known (e.g., certain smoothness assumptions on the objective function), could the results be strengthened to hold for all parameters instead of almost surely?*"
>
> This is unlikely to hold as stated. For instance, if one initializes exactly at an equilibrium, convergence occurs trivially without any constraints. Likewise, if the step sizes happens, by chance, to drive the algorithm to an equilibrium in finitely many iterations, convergence again occurs regardless of any structural assumptions (think about GD on $x \mapsto x^2/2$ with step size 1).
>
> 4. "*Is there a robustness version of the results? For (example) near the limiting point, is the Jacobian’s spectral radius bounded by $1+\delta$ for some small $\delta$ ?*"
>
> Yes because $f$ is assumed to be $C^2$, thus its Jacobian is $C^1$.
>
>  5. "*It would be helpful to include more details about the experimental setup in the appendix. For example, how is the performance of different algorithms compared fairly. Are they run for the same number of epochs or the same number of backpropagations (since the proposed algorithm seems to require more backpropagations per iteration)?*"
>
>  Definitely, we will provide more detail in the appendix. For information we ran the same number of epochs for all algorithms. We did 200 epochs (even for simple task) to make sure that the algorithms converge (or at least, get close to an optimal solution).
>
> [1] Andreyev, A., & Beneventano, P. (2024). Edge of stochastic stability: Revisiting the edge of stability for sgd. arXiv preprint arXiv:2412.20553.
>
> [2] Chemnitz, D., & Engel, M. (2025). Characterizing dynamical stability of stochastic gradient descent in overparameterized learning. Journal of Machine Learning Research, 26(134), 1-46.
>
> [3] Cohen, J. M., Ghorbani, B., Krishnan, S., Agarwal, N., Medapati, S., Badura, M., ... & Gilmer, J. (2022). Adaptive gradient methods at the edge of stability. arXiv preprint arXiv:2207.14484.

---

### Author Response · Authors · 2025-11-18
**General preliminary comment:**

We were very surprised to see several reports claiming that our results bring no genuine novelty, we believe we have failed to emphasize this aspect clearly in our presentation, and we will revise it accordingly. Yet we would like to show, in this preliminary statement, that these claims are inaccurate. To do so, we simply recall the great generality of definability assumptions and present our contributions side-by-side with the state of the art in clear comparison tables:

- our work covers all definable problems, meaning *all known approaches to deep learning we are aware of* (see Remark 2.7). A more extensive illustration can be found in [9] or [10, Appendix A.2].

- to provide a direct, hopefully striking, view of our contribution, we display tables that show in a very clear manner what is gained with respect to the state of the art.

(Note  that extending arguments from quadratic to nonlinear costs is genuinely delicate in dynamical systems, where the theory of linearization is riddled with subtle pitfalls and many counterexamples.)



Lower-bound for the hessian spectrum (avoidance of strict saddle points), assumptions on the loss $f$ and hyperparameters.
| Algorithms            | Our assumptions | Previous works assumptions / results|
| -- |---- |:-- |
| Gradient descent      | $C^2$, definable | $C^2$, $L$-Lipschitz gradient. Step size $\alpha < \frac{1}{L}$ ([1, Theorem 4], [2, Theorem 2]) |
| Heavy ball | $C^2$, definable |$C^2$, $L$-Lipschitz gradient. Step size $\alpha < \frac{1-\beta}{L}$ [3, Lemma 2] or $\alpha < 4/L$ and $\beta$ in some range [8, Theorem 3] |
| Nesterov acceleration | $C^2$, definable | Quadratic [8, Section 4]  |
| SAM| $C^2$, definable | No existing work to our best knowledge   |

Upper-bound for the hessian spectrum (Edge of Stability, EoS), assumptions on the loss $f$ and hyperparameters.
| Algorithms            | Our assumptions  | Previous works assumptions / results |
| --| - |:---- |
| Gradient descent      | $C^2$, definable | Quadratic [4, Proposition 1]. $C^2$, step size $\alpha$ s.t. $\frac{1}{\alpha}$ not an eigenvalue of $\nabla^2 f(x)$ for all critical points $x$, the function $F_\alpha(x) = x - \alpha \nabla f(x)$ preserves Lebesgue null sets by pre-image [5, Theorem 1] |
| Heavy ball| $C^2$, definable | Quadratic [4, Theorem 2])    |
| Nesterov acceleration | $C^2$, definable | Quadratic [4, Theorem 1]  |
| SAM| $C^2$, definable | Overparameterized regression, existence of diverging initializations ([6, Theorem 2.3]). Linear regression, EoS for stochastic SAM [7, Corollary 4.2])                                                                                                              |

[1] Jason D. Lee, Max Simchowitz, Michael I. Jordan, and Benjamin Recht. Gradient descent only converges to minimizers, 29th Annual Conference on Learning Theory.

[2] Ioannis Panageas and Georgios Piliouras. Gradient descent only converges to minimizers: Non-isolated critical points and invariant regions. In 8th Innovations in Theoretical Computer Science Conference.

[3] Tao Sun, Dongsheng Li, Zhe Quan, Hao Jiang, Shengguo Li, and Yong Dou. Heavy-ball algorithms always escape saddle points. In Proceedings of the Twenty-Eighth International Joint Conference on Artificial Intelligence, IJCAI-19.

[4] Jeremy Cohen, Simran Kaur, Yuanzhi Li, J Zico Kolter, and Ameet Talwalkar. Gradient descent on neural networks typically occurs at the edge of stability. In International Conference on Learning Representations, 2021.

[5] Kwangjun Ahn, Jingzhao Zhang, and Suvrit Sra. Understanding the unstable convergence of gradient descent. In International Conference on Machine Learning, pp. 247–257. PMLR, 2022.

[6] Atish Agarwala and Yann Dauphin. Sam operates far from home: eigenvalue regularization as a dynamical phenomenon. In Proceedings of the 40th International Conference on Machine Learning, ICML’23.

[7] Zhanpeng Zhou, Mingze Wang, Yuchen Mao, Bingrui Li, and Junchi Yan. Sharpness-aware minimiza- tion efficiently selects flatter minima late in training. In Proceedings of the International Confer- ence on Learning Representations, ICLR, January 2025.

[8] O’Neill and Wright (2019). Behavior of accelerated gradient methods near critical points of nonconvex functions. Mathematical Programming, 176(1), 403-427.

[9] Jerome Bolte, Edouard Pauwels. A mathematical model for automatic differentiation in machine learning. Conference on Neural Information Processing Systems, Dec 2020, Vancouver, Canada.

[10] Jérôme Bolte, Tam Le, Edouard Pauwels, Antonio Silveti-Falls. Nonsmooth Implicit Differentiation for Machine Learning and Optimization. Advances in Neural Information Processing Systems, Dec 2021

---

### Meta-Review · Area_Chair_yt2J · 2025-12-24

**Summary:**

The paper studies the sharpness properties of points reached by different optimization algorithms, focusing on methods whose iterates are generated as $G^k(x_0)$ for a fixed deterministic update map $G$, which may depend on the hyperparameters. Under structural regularity assumptions on the objective (essentially, definability in an o-minimal structure), the authors show that any convergent point must satisfy that the spectral radius of the Jacobian of $G$ is at most $1$. This condition is interpreted as an eigenvalue-filtering mechanism that rules out convergence to points violating the spectral-radius constraint. Experiments, conducted on small-scale settings (MNIST/Fashion-MNIST and CIFAR-10), are provided to illustrate the theoretical results.

The rebuttal clarified several points raised by the reviewers, but a number of major concerns largely remained. Most importantly, reviewers generally perceived the central spectral-radius result as conceptually well established, in the sense that it generalizes ideas already present in the sharpness and stability literature, albeit previously established in more restricted settings. As a consequence, the contribution was seen by some as either rederiving known results under cleaner assumptions or extending them to regimes of relatively limited relevance to contemporary DL, particularly given the absence of stochasticity and adaptivity.  Although the paper was acknowledged for its rigorous theoretical framework, this reduced enthusiasm for the main results (similar structural assumptions are already used in large parts of optimization and ML theory). In addition, three aspects were indicated as requiring revision and were perceived by some as substantial: first, providing a more thorough comparison and clearer positioning relative to prior work, including more detailed connections to EoS; second, repositioning the proposed algorithms as primarily illustrative tools rather than as practically motivated training methods; and third, an expanded experimental section, updated to reflect current benchmarks.


The authors are encouraged to incorporate the reviewers’ valuable comments and consider resubmission in the future.

**Reviewer Concerns:**

The rebuttal did properly address several issues, for example by clarifying that the results hold under definability and therefore apply more broadly, and by resolving a number of relatively more minor points and misunderstandings.

However, the remaining issues identified in the meta-review appear more substantial in nature.

**Reviewer Scores:**

crYV: likely unchanged (given the reserved tone of the original review, an upward change in score appears unlikely).

UZZ3: unchanged (explicitly stated that they found the required revisions to be substantial and therefore kept the original score).

1fUe: unchanged (still viewed relevance and practical significance as insufficient; kept score, lowered confidence).

ocNd: unchanged (acknowledged clarifications but maintained low perceived practical relevance).

---

### Decision · Program_Chairs · 2026-01-26

Reject